# BiBERT: Accurate Fully Binarized BERT

**Haotong Qin**[*,1,4], **Yifu Ding**[*,1,4], **Mingyuan Zhang**[*2], **Qinghua Yan**[1], **Aishan Liu**[1],
**Qingqing Dang**[3], **Ziwei Liu**[2], **Xianglong Liu**[✉ 1]

[1]State Key Lab of Software Development Environment, Beihang University    [3]Baidu Inc.
[2]S-Lab, Nanyang Technological University    [4]Shen Yuan Honors College, Beihang University
{qinhaotong,yifuding,yanqh,aishanliu,xlliu}@buaa.edu.cn
mingyuan001@e.ntu.edu.sg zwliu.hust@gmail.com dangqingqing@baidu.com

## ABSTRACT

The large pre-trained BERT has achieved remarkable performance on Natural
Language Processing (NLP) tasks but is also computation and memory expen-
sive. As one of the powerful compression approaches, binarization extremely
reduces the computation and memory consumption by utilizing 1-bit parame-
ters and bitwise operations. Unfortunately, the full binarization of BERT (*i.e.*,
1-bit weight, embedding, and activation) usually suffer a significant performance
drop, and there is rare study addressing this problem. In this paper, with the
theoretical justification and empirical analysis, we identify that the severe perfor-
mance drop can be mainly attributed to the information degradation and optimiza-
tion direction mismatch respectively in the forward and backward propagation,
and propose **BiBERT**, an accurate fully binarized BERT, to eliminate the per-
formance bottlenecks. Specifically, BiBERT introduces an efficient *Bi-Attention*
structure for maximizing representation information statistically and a *Direction-
Matching Distillation* (DMD) scheme to optimize the full binarized BERT accu-
rately. Extensive experiments show that BiBERT outperforms both the straight-
forward baseline and existing state-of-the-art quantized BERTs with ultra-low bit
activations by convincing margins on the NLP benchmark. As the first fully bina-
rized BERT, our method yields impressive $56.3\times$ and $31.2\times$ saving on FLOPs and
model size, demonstrating the vast advantages and potential of the fully binarized
BERT model in real-world resource-constrained scenarios.

## 1 INTRODUCTION

Recently, the pre-trained language models have shown great power in
various natural language processing (NLP) tasks (Wang et al., 2018a;
Qin et al., 2019; Rajpurkar et al., 2016). In particular, BERT (De-
vlin et al., 2018) significantly improves the state-of-the-art performance,
while the massive parameters hinder their widespread deployment on
edge devices in the real world. Therefore, model compression has been
actively studied to alleviate resource constraint issues, including quan-
tization (Shen et al., 2020; Zafrir et al., 2019), distillation (Jiao et al.,
2020; Xu et al., 2020), pruning (McCarley et al., 2019; Gordon et al.,
2020), parameter sharing (Lan et al., 2020), *etc.* Among them, quan-
tization emerges as an efficient way to obtain the compact model by
compressing the model parameters to lower bit-width representation,
such as Q-BERT (Shen et al., 2020), Q8BERT (Zafrir et al., 2019), and

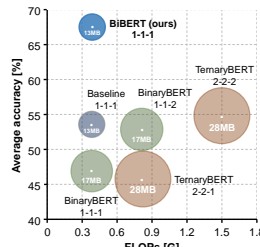

Figure 1: Accuracy vs.
FLOPs & size. Our
BiBERT enjoys most
computation and storage
savings while surpassing
SOTA quantized BERTs
on GLUE benchmark
with low bit activation.

GOBO (Zadeh et al., 2020). However, the representation limitation and optimization difficulties
come as a consequence of applying discrete quantization, triggering severe performance drop in
quantized BERT. Fortunately, distillation becomes a common remedy in quantization as an auxil-
iary optimization approach to tackle the performance drop, which encourages the quantized BERT
to mimic the full-precision model to exploit knowledge in teacher's representation (Bai et al., 2020).

As the quantization scheme with the most aggressive bit-width (Zhou et al., 2016; Wang et al.,
2018b; Xu et al., 2021), full binarization of BERT (*i.e.*, 1-bit weight, word embedding, and ac-

---

[*] equal contribution    [✉] corresponding author

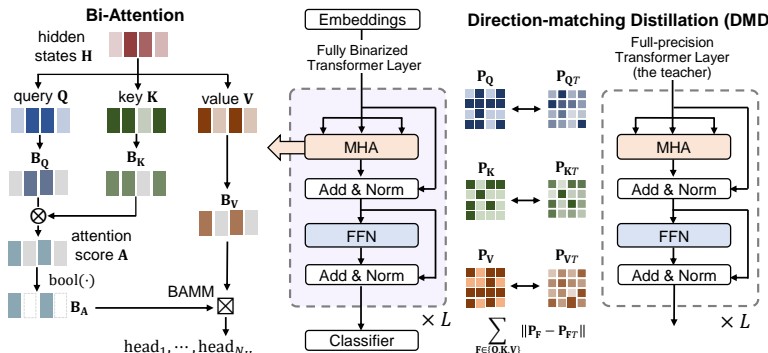

Figure 2: Overview of our BiBERT, applying Bi-Attention structure for maximizing representation information and Direction-Matching Distillation (DMD) scheme for accurate optimization.

tivation) further allows the model to utilize extremely compact 1-bit parameters and efficient bit-wise operations, which can largely promote the applications of BERT on edge devices in the real world (Wang et al., 2021; Liu et al., 2018). However, although the BERT quantization equipped with distillation has been studied, it is still a significant challenge to binarize BERT to extremely 1-bit, especially for its activation. Compared with the weight and word embedding in BERT, the binarization of activation brings the most severe drop, and the model even crashes in NLP tasks. So far, previous studies have pushed down the weight and word embedding to be binarized (Bai et al., 2020), but none of them have ever achieved to binarize BERT with 1-bit activation accurately.

Therefore, we first build a fully binarized BERT baseline, a straightforward yet effective solution based on common techniques. Our study finds that the performance drop of BERT with binarized 1-bit weight, activation, and embedding (called fully binarized BERT) comes from the information degradation of the attention mechanism in the forward propagation and the optimization direction mismatch of distillation in the backward propagation. First, the attention mechanism makes BERT focus selectively on parts of the input and ignore the irrelevant content (Vaswani et al., 2017; Chorowski et al., 2015; Wu et al., 2016). While our analysis shows that direct binarization leads to the almost complete degradation of the information of attention weight (Figure 4), which results in the invalidation of the selection ability for attention mechanism. Second, the distillation for the fully binarized BERT baseline utilizes the attention score, the direct binding product of two binarized activations. However, we show that it causes severe optimization direction mismatch since the non-neglectable error between the defacto and expected optimization direction (Figure 5).

This paper provides empirical observations and theoretical formulations of the above-mentioned phenomena, and proposes a **BiBERT** to turn the full-precision BERT into the strong fully binarized model (see the overview in Figure 2). To tackle the information degradation of attention mechanism, we introduce an efficient *Bi-Attention* structure based on information theory. Bi-Attention applies binarized representations with maximized information entropy, allowing the binarized model to restore the perception of input contents. Moreover, we developed the *Direction-Matching Distillation* (DMD) scheme to eliminate the direction mismatch in distillation. DMD takes appropriate activation and utilizes knowledge from constructed similarity matrices in distillation to optimize accurately.

Our BiBERT, for the first time, presents a promising route towards the accurate fully binarized BERT (with 1-bit weight, embedding, and activation). The extensive experiments on the GLUE (Wang et al., 2018a) benchmark show that our BiBERT outperforms existing quantized BERT models with ultra-lower bit activation by convincing margins. For example, the average accuracy of BiBERT exceeds 1-1-1 bit-width BinaryBERT (1-bit weight, 1-bit embedding and 1-bit quantization) by 20.4% accuracy on average, and even better than 2-8-8 bit-width Q2BERT by 13.3%. Besides, we highlight that our BiBERT gives impressive $\mathbf{56.3\times}$ and $\mathbf{31.2\times}$ saving on FLOPs and model size, respectively, which shows the vast advantages and potential of the fully binarized BERT model in terms of fast inference and flexible deployment in real-world resource-constrained scenarios (Figure 1). Our code is released at https://github.com/htqin/BiBERT.

## 2 BUILDING A FULLY BINARIZED BERT BASELINE

First of all, we build a baseline to study the fully binarized BERT since it has never been proposed in previous works. A straightforward solution is to binarize the representation in BERT architecture in the forward propagation and apply distillation to the optimization in the backward propagation.

## 2.1 BINARIZED BERT ARCHITECTURE

We give a brief introduction to the architecture of binarized BERT. In general, the forward and backward propagation of sign function in binarized network can be formulated as:

$$\text{Forward: } \text{sign}(x) = \begin{cases} 1 & \text{if } x \geq 0 \\ -1 & \text{otherwise} \end{cases}, \qquad \text{Backward: } \frac{\partial C}{\partial x} = \begin{cases} \frac{\partial C}{\partial \, \text{sign}(x)} & \text{if } |x| \leq 1 \\ 0 & \text{otherwise} \end{cases}, \quad (1)$$

where $C$ is the cost function for the minibatch. sign function is applied in the forward propagation while the straight-through estimator (STE) (Bengio et al., 2013) is used to obtain the derivative in the backward propagation. As for the weight of binarized linear layers, the common practice is to redistribute the weight to *zero-mean* for retaining representation information (Rastegari et al., 2016; Qin et al., 2020) and applies scaling factors to minimize quantization errors (Rastegari et al., 2016). The activation is binarized by the sign without re-scaling for computational efficiency. Thus, the computation can be expressed as

$$\text{bi-linear}(\mathbf{X}) = \alpha_{\mathbf{w}}(\text{sign}(\mathbf{X}) \otimes \text{sign}(\mathbf{W} - \mu(\mathbf{W}))), \quad \alpha_{\mathbf{w}} = \frac{1}{n}\|\mathbf{W}\|_{\ell 1}, \qquad (2)$$

where $\mathbf{W}$ and $\mathbf{X}$ denote full-precision weight and activation, $\mu(\cdot)$ denotes the mean value, $\alpha_{\mathbf{w}}$ is the scaling factors for weight, and $\otimes$ denotes the matrix multiplication with bitwise xnor and bitcount as presented in Appendix A.4.

The input data first passes through a binarized embedding layer before being fed into the transformer blocks (Zhang et al., 2020; Bai et al., 2020). And each transformer block consists of two main components: Multi-Head Attention (MHA) module and Feed-Forward Network (FFN). The computation of MHA depends on queries $\mathbf{Q}$, keys $\mathbf{K}$ and values $\mathbf{V}$, which are derived from hidden states $\mathbf{H} \in \mathbb{R}^{N \times D}$. $N$ represents the length of the sequence, and $D$ represents the dimension of features. For a specific transformer layer, the computation in an attention head can be expressed as

$$\mathbf{Q} = \text{bi-linear}_Q(\mathbf{H}), \quad \mathbf{K} = \text{bi-linear}_K(\mathbf{H}), \quad \mathbf{V} = \text{bi-linear}_V(\mathbf{H}), \qquad (3)$$

where $\text{bi-linear}_Q, \text{bi-linear}_K, \text{bi-linear}_V$ represent three different binarized linear layers for $\mathbf{Q}, \mathbf{K}, \mathbf{V}$ respectively. Then we compute the attention score $\mathbf{A}$ as follow:

$$\mathbf{A} = \frac{1}{\sqrt{D}}\left(\mathbf{B_Q} \otimes \mathbf{B_K}^{\top}\right), \quad \mathbf{B_Q} = \text{sign}(\mathbf{Q}), \quad \mathbf{B_K} = \text{sign}(\mathbf{K}), \qquad (4)$$

where $\mathbf{B_Q}$ and $\mathbf{B_K}$ are the binarized query and key, respectively. Note that the obtained attention weight is then truncated by attention mask, and each row in $\mathbf{A}$ can be regarded as a $k$-dim vector, where $k$ is the number of unmasked elements. Then we binarize the attention weights $\mathbf{B_A^s}$ as

$$\mathbf{B_A^s} = \text{sign}(\text{softmax}(\mathbf{A})). \qquad (5)$$

We follow original BERT architecture to carry on the rest of MHA and FFN in the binarized network.

## 2.2 DISTILLATION FOR BINARIZED BERT

Distillation is a common and essential optimization approach to alleviate the performance drop of quantized BERT under ultra-low bit-width settings, which can be unobstructedly applied for any architectures to utilize the knowledge of a full-precision teacher model (Jiao et al., 2020; Bai et al., 2020; Zhang et al., 2020; Wang et al., 2020). The usual practice is to distill the attention score $\mathbf{A}_{Tl}$, MHA output $\mathbf{M}_{Tl}$, and hidden states $\mathbf{H}_{Tl}$ in a layerwise manner from the full-precision teacher network, and transfer to the binarized student counterparts, *i.e.*, $\mathbf{A}_l, \mathbf{M}_l, \mathbf{H}_l$ ($l = 1, ..., L$, where $L$ represents the number of transformer layers), respectively. We use the mean squared errors (MSE) as loss function to measure the difference between student and teacher networks for corresponding features:

$$\ell_{\text{att}} = \sum_{l=1}^{L} \text{MSE}(\mathbf{A}_l, \mathbf{A}_{Tl}), \quad \ell_{\text{mha}} = \sum_{l=1}^{L} \text{MSE}(\mathbf{M}_l, \mathbf{M}_{Tl}), \quad \ell_{\text{hid}} = \sum_{l=1}^{L} \text{MSE}(\mathbf{H}_l, \mathbf{H}_{Tl}). \qquad (6)$$

Then the prediction-layer distillation loss is conducted by minimizing the soft cross-entropy (SCE) between teacher logits $\mathbf{y_T}$ and student logits $\mathbf{y}$. The objective function is expressed as

$$\ell_{\text{distill}} = \ell_{\text{att}} + \ell_{\text{mha}} + \ell_{\text{hid}} + \ell_{\text{pred}}, \qquad \ell_{\text{pred}} = \text{SCE}(\mathbf{y}, \mathbf{y}_T). \qquad (7)$$

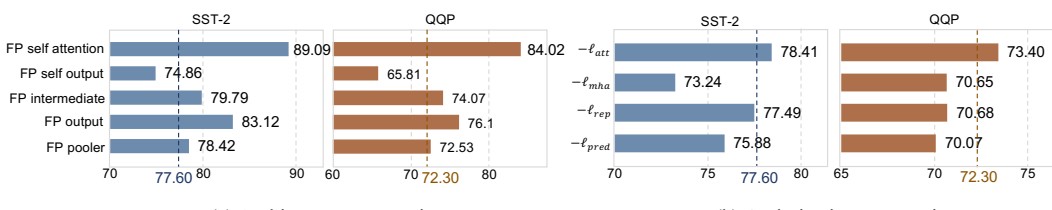

Figure 3: Analysis of bottlenecks from architecture and optimization perspectives. We report the accuracy of binarized BERT on SST-2 and QQP tasks about (a) replace full-precision structure, (b) exclude one distillation knowledge.

## 3 THE RISE OF BiBERT

Although we can build a fully binarized BERT baseline and its training pipeline with common techniques, the performance is still of major concern. Our study shows that the baseline suffers an immense information degradation of attention mechanism in the forward propagation and severe optimization direction mismatch in the backward propagation. To solve these problems, in this section, we propose the BiBERT with theoretical and experimental justifications.

### 3.1 BOTTLENECKS OF FULLY BINARIZED BERT BASELINE

Intuitively, in the fully binarized BERT baseline, the information representation capability largely depends on structures of architecture, such as attention, which is severely limited due to the binarized parameters, and the discrete binarization also makes the optimization more difficult. This means the bottlenecks of the fully binarized BERT baseline come from architecture and optimization for the forward and backward propagation, respectively.

**From the architecture perspective**, we observe the accuracy drop caused by binarizing single structure by replacing it with a full-precision counterpart as Figure 3(a) shows. We find that binarizing MHA brings the most significant drop of accuracy among all parts of the BERT, up to 11.49% (full-precision 89.09% vs. binarized 77.60%) and 11.72% on SST-2 and QQP, respectively. While binarizing FFN and pooler layers brings less harm to the accuracy. Binarization of intermediate or output layer only brings 2.19% and 5.52% drop on SST-2, and binarization of pooler layer only causes 0.82%. Same results can be seen on QQP. Thus, improving the attention structure is the highest priority to solve the accuracy drop of binarized BERT.

**From the optimization perspective**, we eliminate each distillation term to demonstrate the benefits it brings in Figure 3(b). The results show that for most distillation terms, solely removing them in the distillation will harm the performance, *e.g.*, removing the distillation of hidden states leads to 0.11% and 1.62% drop on SST-2 and QQP, and removing that of MHA outputs decrease the model accuracy to 73.24% (4.36% drop) on SST-2. However, when the distillation loss of attention score is removed, the performance increases 0.81% and 1.10% on SST-2 and QQP respectively. These observations inspire us to rethink the distillation of fully binarized BERT. It requires a new design that could utilize the full-precision teacher's knowledge better.

Based on the above experimental observations, we find that (1) the existing attention structure should NOT be directly applied in fully binarized BERT and (2) the distillation for the attention score in fully binarized BERT is actually harmful, which is contrary to the practice of many existing works. Thus in this paper, we first present the theoretical derivation of these two phenomena and then propose a well-designed attention structure and a novel distillation scheme for fully binarized BERT.

### 3.2 BI-ATTENTION

To address the information degradation of binarized representations in the forward propagation, we propose an efficient Bi-Attention structure based on information theory, which statistically maximizes the entropy of representation and revives the attention mechanism in the fully binarized BERT.

#### 3.2.1 INFORMATION DEGRADATION IN ATTENTION STRUCTURE

Since the representations (weight, activation, and embedding) with extremely compressed bit-width in fully binarized BERT have limited capabilities, the ideal binarized representation should preserve

Figure 4: Attention-head view for (a) full-precision BERT, (b) fully binarized BERT baseline, and (c) BiBERT for same input. BiBERT with Bi-Attention shows similar behavior with the full-precision model, while baseline suffers indistinguishable attention for information degradation. The visualization tools is adapted from (Vig, 2019).

the given full-precision counterparts as much as possible, which means the mutual information between binarized and full-precision representations should be maximized. When the deterministic sign function is applied to binarize BERT, the goal is equivalent to maximizing the information entropy $\mathcal{H}(\mathbf{B})$ of binarized representation $\mathbf{B}$ (Messerschmitt, 1971), which is defined as

$$\mathcal{H}(\mathbf{B}) = -\sum_{B} p(B) \log p(B), \tag{8}$$

where $B \in \{-1, 1\}$ is the random variable sampled from $\mathbf{B}$ with probability mass function $p$. Therefore, the information entropy of binarized representation should be maximized to better preserve the full-precision counterparts and let the attention mechanism function well. The application of *zero-mean* pre-binarized weight in binarized linear layers is a representative practice that maximizes the information of binarized weight and activation (as in Section 2.1). The related discussion and proof is shown in Appendix A.1.

As for the attention structure in full-precision BERT, the normalized attention weight obtained by $\mathrm{softmax}$ is essential. But direct application of binarization function causes a complete information loss to binarized attention weight. Specifically, since the $\mathrm{softmax}(\mathbf{A})$ is regarded as following a probability distribution, the elements of $\mathbf{B}_A^s$ are all quantized to 1 (Figure 4(b)) and the information entropy $\mathcal{H}(\mathbf{B}_A^s)$ degenerates to 0. A common measure to alleviate this information degradation is to shift the distribution of input tensors before applying the $\mathrm{sign}$ function, which is formulated as

$$\hat{\mathbf{B}}_{\mathbf{A}}^s = \mathrm{sign}\left(\mathrm{softmax}(\mathbf{A}) - \tau\right), \tag{9}$$

where the shift parameter $\tau$, also regarded as the threshold of binarization, is expected to maximize the entropy of the binarized $\hat{\mathbf{B}}_{\mathbf{A}}^s$ and is fixed during the inference.

**Theorem 1.** *Given* $\mathbf{A} \in \mathbb{R}^k$ *with Gaussian distribution and the variable* $\hat{\mathbf{B}}_{\mathbf{A}}^s$ *generated by* $\hat{\mathbf{B}}_{\mathbf{w}}^A = \mathrm{sign}(\mathrm{softmax}(\mathbf{A}) - \tau)$, *the threshold* $\tau$, *which maximizes the information entropy* $\mathcal{H}(\hat{\mathbf{B}}_{\mathbf{A}}^s)$, *is negatively correlated to the number of elements* $k$.

However, Theorem 1 shows that it is hard to statistically determine $\tau$ to maximize the information entropy of binarized counterparts for the attention weight masked by the changeable length of attention mask. This fact means that common measure for maximizing information (as in binarized linear layers) fails in the binarized attention structure. The proof of Theorem 1 is shown in Appendix A.2.

Moreover, the attention weight obtained by the sign function is binarized to $\{-1, 1\}$, while the original attention weight has a normalized value range $[0, 1]$. The negative value of attention weight in the binarized architecture is contrary to the intuition of the existing attention mechanism and is also empirically proved to be harmful to the attention structure in Appendix C.2.

### 3.2.2 BI-ATTENTION FOR MAXIMUM INFORMATION ENTROPY

To mitigate the information degradation caused by binarization in the attention mechanism, we introduce an efficient Bi-Attention structure for fully binarized BERT, which maximizes information entropy of binarized representations statistically and applies bitwise operations for fast inference.

We first maximize the information entropy $\mathcal{H}(\hat{\mathbf{B}}_{\mathbf{A}}^s)$. The analysis in Section 3.2.1 shows that it is hard to statistically obtain a fixed mean-shift $\tau$ for $\mathrm{softmax}(\mathbf{A})$ that maximizes the entropy of the binarized parameters. Fortunately, since both $\mathrm{softmax}$ and $\mathrm{sign}$ functions are order-preserving, there

is a threshold $\phi(\tau, \mathbf{A})$ to maximize Entropy $\text{sign}(\mathbf{A} - \phi(\tau, \mathbf{A}))$, which is equivalent to maximizing the information entropy of $\hat{\mathbf{B}}_{\mathbf{A}}^s$ in Eq. (9) (as the Proposition 2 proved in the Appendix A.6).

**Theorem 2.** *When the binarized query $\mathbf{B_Q} = \text{sign}(\mathbf{Q}) \in \{-1, 1\}^{N \times D}$ and key $\mathbf{B_K} = \text{sign}(\mathbf{K}) \in \{-1, 1\}^{N \times D}$ are entropy maximized in binarized attention, the probability mass function of each element $\mathbf{A}_{ij}$, $i, j \in [1, N]$ sampled from attention score $\mathbf{A} = \mathbf{B_Q} \otimes \mathbf{B_K}^\top$ can be represented as $p_A(2i - D) = 0.5^D C_D^i$, $i \in [0, D]$, which approximates the Gaussian distribution $\mathcal{N}(0, D)$.*

Theorem 2 shows that the distribution of $\mathbf{A}$ approximates the Gaussian distribution $\mathcal{N}(0, D)$ and is distributed symmetrically, we can thus trivially get that $\phi(\tau, \mathbf{A}) = 0$. The detailed proof is given in Appendix A.3. Therefore, simply binarizing the attention score $\mathbf{A}$ can maximize the information entropy of binarized representation.

Then, to revive the attention mechanism to capture crucial elements, here we are inspired by hard attention (Xu et al., 2015) to binarize the attention weight into the Boolean value, while our design is driven by information entropy maximization. In Bi-Attention, we use $\text{bool}$ function to binarize the attention score $\mathbf{A}$, which is defined as

$$\text{bool}(x) = \begin{cases} 1, & \text{if } x \geq 0 \\ 0, & \text{otherwise} \end{cases}, \qquad \frac{\partial \, \text{bool}(x)}{\partial x} = \begin{cases} 1, & \text{if } |x| \leq 1 \\ 0, & \text{otherwise}. \end{cases} \tag{10}$$

By applying $\text{bool}(\cdot)$ function, the elements in attention weight with lower value are binarized to 0, and thus the obtained entropy-maximized attention weight can filter the crucial part of elements. And the proposed Bi-Attention structure is finally expressed as

$$\mathbf{B_A} = \text{bool}\,(\mathbf{A}) = \text{bool}\left(\frac{1}{\sqrt{D}}\left(\mathbf{B_Q} \otimes \mathbf{B_K}^\top\right)\right), \tag{11}$$

$$\text{Bi-Attention}(\mathbf{B_Q}, \mathbf{B_K}, \mathbf{B_V}) = \mathbf{B_A} \boxtimes \mathbf{B_V}, \tag{12}$$

where $\mathbf{B_V}$ is the binarized value obtained by $\text{sign}(\mathbf{V})$, $\mathbf{B_A}$ is the binarized attention weight, and $\boxtimes$ is a well-designed Bitwise-Affine Matrix Multiplication (BAMM) operator composed by $\otimes$ and $\text{bitshift}$ to align training and inference representations and perform efficient bitwise calculation. The detailed of BAMM is illustrated as Figure 7 in Appendix A.4.

In a nutshell, in our Bi-Attention structure, the information entropy of binarized attention weight is maximized (as Figure 4(c) shows) to alleviate its immense information degradation and revive the attention mechanism. Bi-Attention also achieves greater efficiency since the $\text{softmax}$ is excluded.

## 3.3 DIRECTION-MATCHING DISTILLATION

To address the direction mismatch occurred in fully binarized BERT baseline in the backward propagation, we further propose a Direction-Matching Distillation (DMD) scheme with apposite distilled activations and the well-constructed similarity matrices to effectively utilize knowledge from the teacher, which optimizes the fully binarized BERT more accurately.

### 3.3.1 DIRECTION MISMATCH

As an optimization technique based on element-level comparison of activation, distillation allows the binarized BERT to mimic the full-precision teacher model about intermediate activation. However, we find that the distillation causes direction mismatch for optimization in the fully binarized BERT baseline (Section 3.1), leading to insufficient optimization and even harmful effects.

Eq. (6) shows that, the distillation for attention score $\mathbf{A}$ in a specific layer can be expressed as $\text{MSE}(\mathbf{A}, \mathbf{A}_T)$, where $\mathbf{A}$ and $\mathbf{A}_T$ are attention scores in binarized student BERT and full-precision teacher BERT, respectively. Since the attention score in fully binarized BERT is obtained by multiplying binarized query $\mathbf{B_Q}$ and key $\mathbf{B_K}$, the loss $\ell_{\text{att}}$ can be expressed as:

$$\ell_{\text{att}} = \text{MSE}\left(\frac{1}{\sqrt{D}}\mathbf{B_Q} \otimes \mathbf{B_K}^\top, \frac{1}{\sqrt{D}}\mathbf{Q}_T \times \mathbf{K}_T^\top\right). \tag{13}$$

**Theorem 3.** *Given the variables $X$ and $X_T$ follow $\mathcal{N}(0, \sigma_1), \mathcal{N}(0, \sigma_2)$ respectively, the proportion of optimization direction error is defined as $p_{error\ Q\text{-}bit} = p(\text{sign}(X - X_T) \neq \text{sign}(\text{quantize}_Q(X) - X_T))$, where $\text{quantize}_Q$ denotes the Q-bit symmetric quantization. As Q reduces from 8 to 1, $p_{error\ Q\text{-}bit}$ becomes larger.*

Theorem 4 shows that the distillation for binarized activation based on the numerical comparison with full-precision activation causes the most severe optimization direction error among all bit-widths. For example, for activation following the assumption of standard normal distribution, the error probability caused by 1-bit binarization is approximately $4.4\times$ that of 4-bit quantization (14.4% vs. 3.3%). The sudden increase in the probability of optimization direction mismatch makes it a critical issue in the fully binarized BERT while neglected in the previous BERT quantization studies. The proof and discussion of Theorem 4 is found in Appendix A.5.

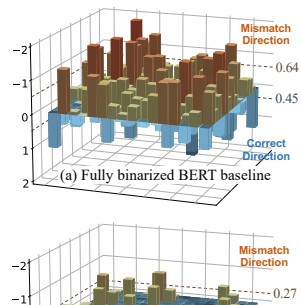

Since the attention score is obtained by direct multiplication of two binarized activations (binarized query $\mathbf{B_Q}$ and key $\mathbf{B_Q}$), its distillation is also misled by the direction mismatch. In Figure 5(a), the direction mismatch in the distillation of attention score is common and severe, even causes up to higher degree optimization on the mismatch direction (mismatch 0.64 vs. match 0.45 on average). The phenomenon in Figure 5 reveals that the element-wise gradient direction error for binarized activation found in Theorem 4 accumulates in the distillation of the attention score and eventually causes a significant mismatch between the direction of overall optimization and that of the distillation loss. Hence, the direction mismatch hinders the accurate optimization of the fully binarized BERT.

Figure 5: Visualization of the direction mismatch of one head throughout training. The full binarized BERT baseline distills the attention score which has severe direction mismatch, while the knowledge (take queries as example in (b)) used in our BiBERT significantly alleviates mismatch.

Second, the activation scales in the fully binarized student and full-precision teacher BERT are significantly different, *e.g.*, $\mathbf{A}$ in fully binarized BERT approximately follows the fixed distribution $\mathcal{N}(0, D)$ while that of $\mathbf{A}_T$ in full-precision teacher is flexible. Applying the discrete binarization function also makes numerical changes of activation more significant. These problems for activation also hinder effective distillation for the fully binarized BERT.

### 3.3.2 DIRECTION-MATCHING DISTILLATION FOR ACCURATE OPTIMIZATION

To solve the optimization direction mismatch in the distillation of the BERT full binarization, we propose the Direction-Matching Distillation (DMD) method in BiBERT.

We first reselect the distilled activations for DMD. As discussed in Section 3.3.1, severe direction mismatch is mainly caused by the distillation of the direct binding product of two binarized activations. Thus, we distill the upstream query $\mathbf{Q}$ and key $\mathbf{K}$ instead of attention score in our DMD for distillation to utilize its knowledge while alleviating direction mismatch. Moreover, inspired by an observation in Section 3.1 that the distillation of MHA output is of great help for improving performance, we also distill the value $\mathbf{V}$ to further cover all the inputs of MHA.

Then, we construct similarity pattern matrices for distilling activation, which can be expressed as

$$\mathbf{P_Q} = \frac{\mathbf{Q} \times \mathbf{Q}^\top}{\|\mathbf{Q} \times \mathbf{Q}^\top\|}, \qquad \mathbf{P_K} = \frac{\mathbf{K} \times \mathbf{K}^\top}{\|\mathbf{K} \times \mathbf{K}^\top\|}, \qquad \mathbf{P_V} = \frac{\mathbf{V} \times \mathbf{V}^\top}{\|\mathbf{V} \times \mathbf{V}^\top\|}, \qquad (14)$$

where $\|\cdot\|$ denotes $\ell 2$ normalization. Previous work shows that matrices constructed in this way are regarded as the specific patterns reflecting the semantic comprehension of network (Tung & Mori, 2019; Martinez et al., 2020). We further find that matrices are also scale-normalized and stable numerically since they focus more on endogenous relative relationships and thus are suitable for distillation between binarized and full-precision networks. The corresponding $\mathbf{P_{Q}}_T, \mathbf{P_{K}}_T, \mathbf{P_{V}}_T$ are constructed in the same way by the teacher's activation. The distillation loss is expressed as:

$$\ell_{\text{distill}} = \ell_{\text{DMD}} + \ell_{\text{hid}} + \ell_{\text{pred}}, \qquad \ell_{\text{DMD}} = \sum_{l \in [1, L]} \sum_{\mathbf{F} \in \mathcal{F}_{\text{DMD}}} \|\mathbf{P_F}l - \mathbf{P_F}_{Tl}\|, \qquad (15)$$

where $L$ denotes the number of transformer layers, $\mathcal{F}_{\text{DMD}} = \{\mathbf{Q}, \mathbf{K}, \mathbf{V}\}$. The loss term $\ell_{\text{hid}}$ is constructed as the $\ell 2$ normalization form, and $\ell_{\text{pred}}$ is still constructed as in Eq. (7).

Our DMD scheme first provides the matching optimization direction (Figure 5(b)) by reselecting appropriate distilled parameters and then constructs similarity matrices to eliminate scale differences and numerical instability, thereby improves fully binarized BERT by accurate optimization.

# 4 EXPERIMENTS

In this section, we conduct extensive experiments to validate the effectiveness of our proposed BiB-ERT for efficient learning on the multiple architectures and the GLUE (Wang et al., 2018a) benchmark with diverse NLP tasks. We first conduct an ablation study on four tasks (SST-2, MRPC, RTE, and QQP) for BiBERT on BERT$_{BASE}$ (12 hidden layers) architecture (Devlin et al., 2019) to showcase the benefits of the Bi-Attention structure and DMD scheme separately. Then we compare BiBERT with the state-of-the-art (SOTA) quantized BERTs in terms of accuracy on BERT$_{BASE}$. Our designs stand out among fully binarized BERTs and even outperform some quantized models with more bit-width parameters. We also evaluate our BiBERT on TinyBERT$_{6L}$ (6 hidden layers) and TinyBERT$_{4L}$ (4 hidden layers) (Jiao et al., 2020), and BiBERT on these compact architectures even outperforms existing methods on BERT$_{BASE}$. In terms of efficiency, our BiBERT achieves an impressive $56.3\times$ and $31.2\times$ saving on FLOPs and model size. The detailed experimental setup and implementations are given in Appendix B.

## 4.1 ABLATION STUDY

As shown in Table 1, the fully binarized BERT baseline suffers a severe performance drop on SST-2, MRPC, RTE and QQP tasks. Bi-Attention and DMD can improve the performance when used alone, and the two techniques further boost the performance considerably when combined together. To conclude, the two techniques can promote each other to improve BiB-ERT and close the performance gap between fully binarized BERT and full-precision counterpart.

Table 1: Ablation study.

| Quant | #Bits | DA | SST-2 | MRPC | RTE | QQP |
|---|---|---|---|---|---|---|
| Full Precision | 32-32-32 | – | 93.2 | 86.3 | 72.2 | 91.4 |
| Baseline | 1-1-1 | ✗ | 77.6 | 70.2 | 54.1 | 73.2 |
| Bi-Attention | 1-1-1 | ✗ | 82.1 | 70.5 | 55.6 | 74.9 |
| DMD | 1-1-1 | ✗ | 79.9 | 70.5 | 55.2 | 75.3 |
| BiBERT (ours) | 1-1-1 | ✗ | **88.7** | **72.5** | **57.4** | **84.8** |
| Baseline | 1-1-1 | ✓ | 84.0 | 71.4 | 50.9 | - |
| Bi-Attention | 1-1-1 | ✓ | 85.6 | 73.2 | 53.1 | - |
| DMD | 1-1-1 | ✓ | 85.3 | 72.5 | 56.3 | - |
| BiBERT (ours) | 1-1-1 | ✓ | **90.9** | **78.8** | **61.0** | - |

## 4.2 COMPARISON WITH SOTA METHODS

We compare our BiBERT with the SOTA BERT quantization methods under ultra-low bit-width in terms of accuracy and efficiency, to fully demonstrate the advantages of our design.

**Accuracy Performance.** In Table 2 and Table 3, we show experiments on the BERT$_{BASE}$ (as default), TinyBERT$_{6L}$, and TinyBERT$_{4L}$ architectures and the GLUE benchmark with or without data augmentation. Since the MNLI and QQP have large data volume and it brings little benefits for the performance to apply data augmentation, we do not apply augmentation for these two tasks, thus they are excluded in Table 3. Results show that BiBERT outperforms other methods on the development set of GLUE benchmark, including TernaryBERT, BinaryBERT, Q-BERT, and Q2BERT.

Table 2 shows the results on the GLUE benchmark without data augmentation. Our BiBERT surpasses existing methods on BERT$_{BASE}$ architecture by a wide margin in the average accuracy, and first achieves convergence under ultra-low bit activation on some tasks, such as CoLA, MRPC, and RTE. While under ultra-low bit activation, the accuracy of TernaryBERT and BinaryBERT decreases severely which also does not improve under the 2-2-2 setting (about $4\times$ FLOPs and $2\times$ storage usage increase). On some specific tasks like MRPC, higher bit-widths and consumption did not even make TernaryBERT and BinaryBERT converge. In addition, BiBERT surpasses the fully binarized baseline$_{50\%}$ and BinaryBERT$_{50\%}$ (maximizing the information entropy by the 50% quantile threshold) and other strengthened baselines, we present detailed discussion in Appendix C.1. With data augmentation, BiBERT achieves comparable performances with full-precision BERT on several tasks, *e.g.*, 90.9% accuracy (drop only 2.2%) on SST-2 (Table 3). These results indicate that BiBERT makes full use of the limited representation capabilities by the well-designed structure and training scheme. We noticed that BiBERT lags behind Q-BERT (2-8-8) and TernaryBERT (2-2-2) on MNLI and STS-B (with DA) while surpassing them on other tasks, indicating the fully binarized BERT may have greater improvement potential in these tasks. BiBERT is also evaluated on compact TinyBERT$_{6L}$ and TinyBERT$_{4L}$ architectures, with about $2.0\times$ and $18.8\times$ fewer FLOPs, respectively. The results show that BiBERT on these compact architectures still outperforms existing quantization methods on BERT$_{BASE}$, such as TernaryBERT and BinaryBERT. It forcefully demonstrates that the targeted designs of BiBERT enable fully binarized BERTs to run on various architectures.

---

"#Bits" (W-E-A) is the bit number for weights, word embedding, and activations. "DA" is short for data augmentation. "Avg." denotes the average results.

Table 2: Comparison of BERT quantization methods without data augmentation.

| Quant | #Bits | Size (MB) | FLOPs (G) | MNLI-m/mm | QQP | QNLI | SST-2 | CoLA | STS-B | MRPC | RTE | Avg. |
|---|---|---|---|---|---|---|---|---|---|---|---|---|
| Full Precision | 32-32-32 | 418 | 22.5 | 84.9/85.5 | 91.4 | 92.1 | 93.2 | 59.7 | 90.1 | 86.3 | 72.2 | 83.9 |
| Q-BERT | 2-8-8 | 43.0 | 6.5 | 76.6/77.0 | – | – | 84.6 | – | – | 68.3 | 52.7 | – |
| Q2BERT | 2-8-8 | 43.0 | 6.5 | 47.2/47.3 | 67.0 | 61.3 | 80.6 | – | 4.4 | 68.4 | 52.7 | 47.7 |
| TernaryBERT | 2-2-8 | 28.0 | 6.4 | 83.3/83.3 | 90.1 | – | – | 50.7 | – | 87.5 | 68.2 | – |
| BinaryBERT | 1-1-4 | 16.5 | 1.5 | 83.9/84.2 | 91.2 | 90.9 | 92.3 | 44.4 | 87.2 | 83.3 | 65.3 | 79.9 |
| TernaryBERT | 2-2-2 | 28.0 | 1.5 | 40.3/40.0 | 63.1 | 50.0 | 80.7 | 0 | 12.4 | 68.3 | 54.5 | 45.5 |
| BinaryBERT | 1-1-2 | 16.5 | 0.8 | 62.7/63.9 | 79.9 | 52.6 | 82.5 | 14.6 | 6.5 | 68.3 | 52.7 | 53.7 |
| TernaryBERT | 2-2-1 | 28.0 | 0.8 | 32.7/33.0 | 74.1 | 59.3 | 53.1 | 0 | 7.1 | 68.3 | 53.4 | 42.3 |
| Baseline | 1-1-1 | 13.4 | 0.4 | 45.8/47.0 | 73.2 | 66.4 | 77.6 | 11.7 | 7.6 | 70.2 | 54.1 | 50.4 |
| Baseline$_{50\%}$ | 1-1-1 | 13.4 | 0.4 | 47.7/49.1 | 74.1 | 67.9 | 80.0 | 14.0 | 11.5 | 69.8 | 54.5 | 52.1 |
| BinaryBERT | 1-1-1 | 16.5 | 0.4 | 35.6/35.3 | 66.2 | 51.5 | 53.2 | 0 | 6.1 | 68.3 | 52.7 | 41.0 |
| BinaryBERT$_{50\%}$ | 1-1-1 | 13.4 | 0.4 | 39.2/40.0 | 66.7 | 59.5 | 54.1 | 4.3 | 6.8 | 68.3 | 53.4 | 43.5 |
| **BiBERT (ours)** | **1-1-1** | **13.4** | **0.4** | **66.1/67.5** | **84.8** | **72.6** | **88.7** | **25.4** | **33.6** | **72.5** | **57.4** | **63.2** |
| Full Precision $_{6L}$ | 32-32-32 | 257 | 11.3 | 84.6/83.2 | 71.6 | 90.4 | 93.1 | 51.1 | 83.7 | 87.3 | 70.0 | 79.4 |
| **BiBERT$_{6L}$ (ours)** | **1-1-1** | **6.8** | **0.2** | **63.6/63.7** | **83.3** | **73.6** | **87.9** | **24.8** | **33.7** | **72.2** | **55.9** | **62.1** |
| Full Precision $_{4L}$ | 32-32-32 | 55.6 | 1.2 | 82.5/81.8 | 71.3 | 87.7 | 92.6 | 44.1 | 80.4 | 86.4 | 66.6 | 77.0 |
| **BiBERT$_{4L}$ (ours)** | **1-1-1** | **4.4** | **0.03** | **55.3/56.1** | **78.2** | **71.2** | **85.4** | **14.9** | **31.5** | **72.2** | **54.2** | **57.7** |

Table 3: Comparison of BERT quantization methods with data augmentation.

| Quant | #Bits | Size (MB) | FLOPs (G) | QNLI | SST-2 | CoLA | STS-B | MRPC | RTE | Avg. |
|---|---|---|---|---|---|---|---|---|---|---|---|
| Full Precision | 32-32-32 | 418 | 22.5 | 92.1 | 93.2 | 59.7 | 90.1 | 86.3 | 72.2 | 82.3 |
| TernaryBERT | 2-2-8 | 28.0 | 6.4 | 90.0 | 92.9 | 47.8 | 84.3 | 82.6 | 68.4 | 77.8 |
| BinaryBERT | 1-1-4 | 16.5 | 1.5 | 91.4 | 93.7 | 53.3 | 88.6 | 86.0 | 71.5 | 80.8 |
| TernaryBERT | 2-2-2 | 28.0 | 1.5 | 50.0 | 87.5 | 20.6 | 72.5 | 72.0 | 47.2 | 58.3 |
| BinaryBERT | 1-1-2 | 16.5 | 0.8 | 51.0 | 89.6 | 33.0 | 11.4 | 71.0 | 55.9 | 52.0 |
| TernaryBERT | 2-2-1 | 28.0 | 0.8 | 50.9 | 80.3 | 6.5 | 10.3 | 71.5 | 53.4 | 45.5 |
| Baseline | 1-1-1 | 13.4 | 0.4 | 69.2 | 84.0 | 23.3 | 14.4 | 71.4 | 50.9 | 52.2 |
| BinaryBERT | 1-1-1 | 16.5 | 0.4 | 66.1 | 78.3 | 7.3 | 22.1 | 69.3 | 57.7 | 50.1 |
| **BiBERT (ours)** | **1-1-1** | **13.4** | **0.4** | **76.0** | **90.9** | **37.8** | **56.7** | **78.8** | **61.0** | **67.0** |
| Full Precision $_{6L}$ | 32-32-32 | 257 | 11.3 | 90.4 | 93.1 | 51.1 | 83.7 | 87.3 | 70.0 | 79.2 |
| **BiBERT$_{6L}$ (ours)** | **1-1-1** | **6.8** | **0.2** | **76.0** | **90.7** | **35.6** | **62.7** | **77.9** | **57.4** | **66.7** |
| Full Precision $_{4L}$ | 32-32-32 | 55.6 | 1.2 | 87.7 | 92.6 | 44.1 | 80.4 | 86.4 | 66.6 | 76.2 |
| **BiBERT$_{4L}$ (ours)** | **1-1-1** | **4.4** | **0.03** | **73.2** | **88.3** | **20.0** | **42.5** | **74.0** | **56.7** | **59.1** |

**Efficiency Performance.** As shown in Table 2 and Table 3 above, our BiBERT achieves an impressive $56.3\times$ FLOPs and $31.2\times$ model size saving over the full-precision BERT. Furthermore, benefiting from simple yet effective Bi-Attention structure which casts the expensive softmax operation into a well-engineered bit operation BAMM, our BiBERT surpasses other quantized BERTs in computation and storage saving while enjoying the best accuracy.

## 4.3 MORE ANALYSIS

**Information Performance.** To show the improvement of information performance by applying Bi-Attention, we compare the information entropy of binarized representations for baseline and BiBERT. As shown in Figure 6(a), we take the first heads in layer 0 of each model, and the same phenomenon exists in all heads and layers. During the training process, the information entropy of attention weight in BiBERT fluctuates in a small range and is almost maximized, however, that of baseline is completely degraded to 0.

**Training Curves.** We plot training loss curves of fully binarized BERT baseline and BiBERT on SST-2 without data augmentation in Figure 6(b). Compared with the baseline model, our method has a faster convergence rate and achieves higher accuracy, suggesting ours advantages in terms of accurate optimization.

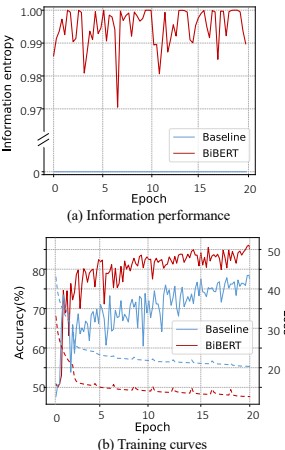

(a) Information performance

(b) Training curves

Figure 6: Analysis.

## 5 CONCLUSION

We propose BiBERT towards the accurate fully binarized BERT. We first reveal the bottlenecks of the fully binarized BERT baseline and build a theoretical foundation for the impact of full binarization. Then we propose Bi-Attention and DMD in BiBERT to improve performance. BiBERT outperforms existing SOTA BERT quantization methods with ultra-low bit activation, giving an impressive $56.3\times$ FLOPs and $31.2\times$ model size saving. Our work gives an insightful analysis and effective solution about the crucial issues in BERT full binarization, which blazes a promising path for the extreme compression of BERT. We hope our work can provide directions for future research.

**Acknowledgement** This work was supported in part by National Key Research and Development Plan of China under Grant 2020AAA0103503, National Natural Science Foundation of China under Grant 62022009 and Grant 61872021, Beijing Nova Program of Science and Technology under Grant Z191100001119050, Outstanding Research Project of Shen Yuan Honors College, BUAA Grant 230121206, and CCF-Baidu Open Fund OF2021003; it was also supported under the RIE2020 Industry Alignment Fund-Industry Collaboration Projects (IAF-ICP) Funding Initiative.

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

# Appendix for BiBERT

## A    Main Proofs and Discussion

### A.1    Discussion of Binarized Linear Layer

**Proposition 1.** *When a random variable $X$ follows a* zero-mean *Gaussian distribution, the information entropy $\mathcal{H}(B)$ is maximized, where $B_X = \text{sign}(X)$.*

Since the value of $\mathbf{B_w}$ depends on the sign of $\mathbf{W}$ and the distribution of $\mathbf{W}$ is almost symmetric (He & Fan, 2019; Banner et al., 2018), the balanced operation can maximize the information entropy of binarized $\mathbf{B_w}$ on the whole. The balanced binarized weights with the zero-mean attribute can be obtained by subtracting the mean of full-precision weights. As Proposition 1 shown, under the binomial distribution assumption and symmetric assumption of $\mathbf{W}$, when the binarized weight is balanced, the information entropy of $\mathbf{B_w}$ takes the maximum value, which means the binarized values should be evenly distributed.

Moreover, when the weight is zero-mean, the entropy of output $\mathbf{Z}$ (seen as the activation in the next binarized linear layer) in the network can also be maximized. Supposing quantized activations $\mathbf{B_x}$ have mean $\mathbb{E}[\mathbf{B_x}] = \mu \mathbf{1}$, the mean of $\mathbf{Z}$ can be calculated by

$$\mathbb{E}[\mathbf{Z}] = \mathbf{B_w} \otimes \mathbb{E}[\mathbf{B_x}] = \mathbf{B_w} \otimes \mu \mathbf{1}. \tag{16}$$

Since the zero-mean weight is applied in each layer, we have $\mathbf{Q_w} \otimes \mathbf{1} = 0$, and the mean of output is zero. Therefore, the information entropy of activations in each layer can be maximized according to Proposition 1. Therefore, a simple redistribution for the full-precision counterpart of binarized weights can simultaneously maximize the information entropy of binarized weights and activations.

The proof of Proposition 1 is presented as below:

*Proof.* According to the definition of information entropy, we have

$$\mathcal{H}(B_X) = - \sum_{b \in \mathcal{B}_{\mathcal{X}}} p_{B_X}(b) \log p_{B_X}(b) \tag{17}$$

$$= - p_{B_X}(-1) \log p_{B_X}(-1) - p_{B_X}(1) \log p_{B_X}(1) \tag{18}$$

$$= - p_{B_X}(-1) \log p_{B_X}(-1) - (1 - p_{B_X}(-1) \log (1 - p_{B_X}(-1))). \tag{19}$$

Then we can get the derivative of $\mathcal{H}(B_X)$ with respect to $p_{B_X}(-1)$

$$\frac{d\,\mathcal{H}(B_X)}{d\,p_{B_X}(-1)} = - \left( \log p_{B_X}(-1) + \frac{p_{B_X}(-1)}{p_{B_X}(-1)\ln 2} \right) + \left( \log (1 - p_{B_X}(-1)) + \frac{1 - p_{B_X}(-1)}{(1 - p_{B_X}(-1))\ln 2} \right) \tag{20}$$

$$= - \log p_{B_X}(-1) + \log (1 - p_{B_X}(-1)) - \frac{1}{\ln 2} + \frac{1}{\ln 2} \tag{21}$$

$$= \log \left( \frac{1 - p_B(-1)}{p_B(-1)} \right). \tag{22}$$

When we let $\frac{d\,\mathcal{H}(B_X)}{d\,p_{B_X}(-1)} = 0$ to maximize the $\mathcal{H}(B_X)$, we have $p_{B_X}(-1) = 0.5$. Since the deterministic sign function with the *zero* threshold is applied as the quantizer, the probability mass function of $B_X$ is represented as

$$p_{B_X}(b) = \begin{cases} \int_{-\infty}^{0} f_Y(y)\,dy, & \text{if } b = -1 \\ \int_{0}^{\infty} f_Y(y)\,dy, & \text{if } b = 1, \end{cases} \tag{23}$$

where $f_X(x)$ is the probability density function of variable $X$. Since $X \sim \mathcal{N}(0, \sigma)$, the $f_X(x)$ is defined as

$$f_X(x) = \frac{1}{\sigma\sqrt{2\pi}} e^{-\frac{x^2}{2\sigma^2}}. \tag{24}$$

When the information entropy $\mathcal{H}(B_X)$ is maximized, we have

$$\int_{-\infty}^{0} f_X(x)\,dx = 0.5. \tag{25}$$

$\square$

## A.2 Proof of the Theorem 1

**Theorem 1.** *Given $\mathbf{A} \in \mathbb{R}^k$ with Gaussian distribution and the variable $\hat{\mathbf{B}}_{\mathbf{A}}^s$ generated by $\hat{\mathbf{B}}_{\mathbf{w}}^A = \mathrm{sign}(\mathrm{softmax}(\mathbf{A}) - \tau)$, the threshold $\tau$, which maximizes the information entropy $\mathcal{H}(\hat{\mathbf{B}}_{\mathbf{A}}^s)$, is negatively correlated to the number of elements $k$.*

*Proof.* Given the $\mathbf{A} = \{A_1, A_2, ..., A_k\} \in \mathbb{R}^k$, each $A_i$ obeys Gaussian distribution $\mathcal{N}(\mu, \sigma)$. Without loss of generality, we consider the threshold of the first variable $A_1$, which can maximize the information entropy. Such threshold $\tau_K$ satisfies

$$\underbrace{\int_{-\infty}^{\infty} \cdots \int_{-\infty}^{\infty}}_{K-1} \int_{-\infty}^{\infty} \prod_{i=1}^{K} p_{A_i}(a_i) [\mathrm{softmax}_K(a_1) \leq \tau_K]\,da_i = 0.5, \tag{26}$$

where $[\cdot]$ denotes the $Iverson\ bracket$ that is defined as

$$[P] = \begin{cases} 1 & \text{if } P \text{ is true;} \\ 0 & \text{otherwise,} \end{cases} \tag{27}$$

and the softmax function is

$$\mathrm{softmax}_k(A_i) = \frac{e^{A_i}}{\sum_{j=1}^{k} e^{A_j}}. \tag{28}$$

$p_{A_i}(a_i)$ presents the probability that the $i$-th element is equal to $a_i$. Since the $\mathrm{softmax}_k$ function is order-preserving, there exists exactly one threshold $\tau(A_2, A_3, ..., A_k; \tau_k)$ such that $e^{\tau(A_2, A_3, ..., A_k; \tau_k)} / (e^{\tau(A_2, A_3, ..., A_k; \tau_k)} + \sum_{j=2}^{K} e^{a_j}) = \tau_k$.

In other words, we can convert the after-softmax threshold $\tau_k$ to a before-softmax threshold $\tau(A_2, A_3, ..., A_k; \tau_k)$ for $A_1$. Then the Eq. (26) can be express as

$$\underbrace{\int_{-\infty}^{\infty} \cdots \int_{-\infty}^{\infty}}_{k-1} \int_{-\infty}^{\tau^s(k)} \prod_{i=1}^{k} p_{A_i}(a_i)\,da_i = 0.5. \tag{29}$$

When we takes the $\{k+1\}$-th variable $A_{k+1}$ into consider. Since the $\sum_{i=2}^{k} e^{A_i} < \sum_{i=2}^{k+1} e^{A_i}$ is always satisfied, $\tau(A_2, A_3, ..., A_k, A_{k+1}; \tau_k) > \tau(A_2, A_3, ..., A_k; \tau_k)$ is also always satisfied. Consider the function $F(x)$, which is defined below:

$$F(x) = \underbrace{\int_{-\infty}^{\infty} \cdots \int_{-\infty}^{\infty}}_{k-1} \int_{-\infty}^{x} \prod_{i=1}^{k} p_{A_i}(a_i)\, da_i \tag{30}$$

$F(x)$ is a strictly monotone increasing function, which means $F(\tau(A_2, A_3, ..., A_k, A_{k+1}; \tau_k)) > F(\tau(A_2, A_3, ..., A_k; \tau_k))$.

Then we have

$$\underbrace{\int_{-\infty}^{\infty} \int_{-\infty}^{\infty} \cdots \int_{-\infty}^{\infty}}_{k} \int_{-\infty}^{\tau(A_2, A_3, ..., A_k, A_{k+1}; \tau_k)} \prod_{i=1}^{k+1} p_{A_i}(a_i)\, da_i. \tag{31}$$

$$> \underbrace{\int_{-\infty}^{\infty} \cdots \int_{-\infty}^{\infty}}_{k-1} \int_{-\infty}^{\infty} \int_{-\infty}^{\tau(A_2, A_3, ..., A_k; \tau_k)} \prod_{i=1}^{k+1} p_{A_i}(a_i)\, da_i \tag{32}$$

$$= \underbrace{\int_{-\infty}^{\infty} \cdots \int_{-\infty}^{\infty}}_{k-1} \int_{-\infty}^{\tau(A_2, A_3, ..., A_k; \tau_k)} \prod_{i=1}^{k} p_{A_i}(a_i)\, da_i \tag{33}$$

$$= 0.5. \tag{34}$$

Since the information entropy of $\mathrm{sign}(\mathrm{softmax}_{K+1}(A_1 - \tau_{K+1}))$ is maximized, we have

$$\underbrace{\int_{-\infty}^{\infty} \int_{-\infty}^{\infty} \cdots \int_{-\infty}^{\infty}}_{k} \int_{-\infty}^{\tau(A_2, A_3, ..., A_k, A_{k+1}; \tau_{k+1})} \prod_{i=1}^{k+1} p_{A_i}(a_i)\, da_i = 0.5, \tag{35}$$

thus,

$$\underbrace{\int_{-\infty}^{\infty} \int_{-\infty}^{\infty} \cdots \int_{-\infty}^{\infty}}_{k} \int_{-\infty}^{\tau(A_2, A_3, ..., A_k, A_{k+1}; \tau_{k+1})} \prod_{i=1}^{k+1} p_{A_i}(a_i)\, da_i \tag{36}$$

$$< \underbrace{\int_{-\infty}^{\infty} \int_{-\infty}^{\infty} \cdots \int_{-\infty}^{\infty}}_{k} \int_{-\infty}^{\tau(A_2, A_3, ..., A_k, A_{k+1}; \tau_k)} \prod_{i=1}^{k+1} p_{A_i}(a_i)\, da_i. \tag{37}$$

Then we can get $\tau_{k+1} < \tau_k$. Therefore, the threshold $\tau$ which maximizes the information entropy $\mathcal{H}(\hat{\mathbf{B}}_{\mathbf{A}}^s)$ is negatively correlated to the number of elements $k$.

$\square$

### A.3 PROOF OF THE THEOREM 2

**Theorem 2.** *When the binarized query $\mathbf{B_Q} = \mathrm{sign}(\mathbf{Q}) \in \{-1, 1\}^{N \times D}$ and key $\mathbf{B_K} = \mathrm{sign}(\mathbf{K}) \in \{-1, 1\}^{N \times D}$ are entropy maximized in binarized attention, the probability mass function of each element $\mathbf{A}_{ij}$, $i, j \in [1, N]$ sampled from attention score $\mathbf{A} = \mathbf{B_Q} \otimes \mathbf{B_K}^{\top}$ can be represented as $p_A(2i - D) = 0.5^D C_D^i$, $i \in [0, D]$, which approximates the Gaussian distribution $\mathcal{N}(0, D)$.*

*Proof.* First, we prove that the distribution of $A_{ij}$ can be approximated as a normal distribution. Considering the definition of $A$, $A_{ij}$ can be expressed as

$$A_{ij} = \sum_{l=1}^{D} B_{\mathbf{Q},il} \times B_{\mathbf{K},jl},$$

where $B_{\mathbf{Q},il}$ represents the $j$-th element of $i$-th vector of $B_{\mathbf{Q}}$ and $B_{\mathbf{K},il}$ represents the $j$-th element of $i$-th vector of $B_{\mathbf{K}}$. The value of the element $B_{\mathbf{Q},il} \times B_{\mathbf{K},jl}$ can be expressed as

$$B_{\mathbf{Q},il} \times B_{\mathbf{K},jl} = \begin{cases} 1, & \text{if } B_{\mathbf{Q},il} \veebar B_{\mathbf{K},jl} = 1 \\ -1, & \text{if } B_{\mathbf{Q},il} \veebar B_{\mathbf{K},jl} = -1. \end{cases} \tag{38}$$

The $B_{\mathbf{Q},il} \times B_{\mathbf{K},jl}$ only can take from two values and its value can be considered as the result of one Bernoulli trial. Thus for the random variable $A_{ij}$ sampled from the output tensor $A$, the probability mass function, $p_A$ can be expressed as

$$p_A(2i - D) = C_D^i \, p_e^i (1 - p_e)^{D-i}, \tag{39}$$

where $p_e$ denotes the probability that the element $B_{\mathbf{Q},il} \times B_{\mathbf{K},jl}$ takes 1. According to the *De Moivre–Laplace* theorem (Walker & Helen, 1985), the normal distribution $\mathcal{N}(\mu, \sigma^2)$ can be used as an approximation of the binomial distribution under certain conditions, and the $p_A(2i - D)$ can be approximated as

$$p_A(2i - D) = C_D^i \, p_e^i (1 - p_e)^{D-i} \simeq \frac{1}{\sqrt{2\pi D p_e (1 - p_e)}} e^{-\frac{(i - D p_e)^2}{2 D p_e (1 - p_e)}}, \tag{40}$$

and then, we can get the mean $\mu = 0$ and variance $\sigma = \sqrt{D}$ of the approximated distribution $\mathcal{N}$. Now we give proof of this below.

According to Proposition 1, both $B_{\mathbf{Q},il}$ and $B_{\mathbf{K},il}$ have equal probability to be 1 or $-1$, which means $p_e = 0.5$. Then we can rewrite the equation as

$$p_A(2i - D) = 0.5^D C_D^i, i \in \{0, 1, 2, ..., D\}. \tag{41}$$

Then we move to calculate the mean and standard variation of this distribution. The mean of this distribution is defined as

$$\mu(p_A) = \sum (2i - D) 0.5^D C_D^i, i \in \{0, 1, 2, ..., D\}. \tag{42}$$

By the virtue of binomial coefficient, we have

$$(2i - D) 0.5^D C_D^i + (2(D - i) - D) 0.5^D C_D^{D-i} = 0.5^D ((2i - D) C_D^i + (D - 2i) C_D^{D-i}) \tag{43}$$
$$= 0.5^D ((2i - D) C_D^i + (D - 2i) C_D^i) \tag{44}$$
$$= 0. \tag{45}$$

Besides, when $D$ is an even number, we have $(2i - D) 0.5^D C_D^i = 0, i = \frac{D}{2}$. These equations prove the symmetry of function $(2i - D) 0.5^D C_D^i$. Finally, we have

$$\mu(p_A) = \sum (2i - D) 0.5^D C_D^i, i \in \{0, 1, 2, ..., D\} \tag{46}$$

$$= \sum ((2i - D) 0.5^D C_D^i + (2(D - i) - D) 0.5^D C_D^{D-i}), i \in \{0, 1, 2, ..., \frac{D}{2}\} \tag{47}$$

$$= 0. \tag{48}$$

The standard variation of $p_A$ is defined as

$$\sigma(p_A) = \sqrt{\left(\sum |2i - D|^2 0.5^D C_D^i\right)} \tag{49}$$

$$= \sqrt{\sum \left(4i^2 - 4iD + D^2\right) 0.5^D C_D^i} \tag{50}$$

$$= \sqrt{0.5^D \left(4 \sum i^2 C_D^i - 4D \sum i C_D^i + D^2 \sum C_D^i\right)}. \tag{51}$$

To calculate the standard variation of $p_Z$, we use Binomial Theorem and have several identical equations:

$$\sum C_D^i = (1+1)^D = 2^D \tag{52}$$

$$\sum i C_D^i = D(1+1)^{D-1} = D2^{D-1} \tag{53}$$

$$\sum i^2 C_D^i = D(D+1)(1+1)^{D-2} = D(D+1)D2^{D-2}. \tag{54}$$

These identical equations help simplify Eq. (51):

$$\sigma(p_A) = \sqrt{0.5^D \left(4 \sum i^2 C_D^i - 4D \sum i C_D^i + D^2 \sum C_D^i\right)} \tag{55}$$

$$= \sqrt{0.5^D (4D(D+1)2^{D-2} - 4D^2 2^{D-1} + D^2 2^D)} \tag{56}$$

$$= \sqrt{0.5^D ((D^2 + D)2^D - 2D^2 2^D + D^2 2^D)} \tag{57}$$

$$= \sqrt{0.5^D (D2^D)} \tag{58}$$

$$= \sqrt{D}. \tag{59}$$

Now we proved that, the distribution of output is approximate normal distribution $\mathcal{N}(0, D)$. $\qquad\square$

### A.4 BAMM OPERATION IN BI-ATTENTION

In the Bi-Attention structure, we apply the $\mathrm{bool}$ function to obtain the binarized attention weights $\mathbf{B}$ with values of 0 and 1, which maximizes the information entropy while restoring the perception of attention mechanism for the input. However, we noticed that in the actual hardware deployment, the 1-bit matrix stored in the hardware is unified into the same form (with binary values of 1 and -1) and is supported by most existing hardware.

Therefore, we propose a new bitwise operation $\boxtimes$ to support the computation between the binarized attention weight $\mathrm{bool}(\mathbf{A})$ and the binarized value $\mathbf{B_V}$ during inference, which is defined as

$$\mathrm{bool}(\mathbf{A}) \boxtimes \mathbf{B_V} = \left(\mathbf{B_A}' \otimes \mathbf{B_V} + \mathbf{B_A}' \otimes \mathbf{1}\right) \gg 1 \tag{60}$$

where $\mathbf{B_A}' \in \{-1, 1\}^{N \times D}$ is the representation of $\mathrm{bool}(\mathbf{A})$ on hardware that $\mathbf{B_A}'_{ij} = -1$ where $\mathrm{bool}(\mathbf{A})_{ij} = 0$, $\mathbf{1} \in \{1\}^{N \times D}$ is an all ones matrix, and $\otimes$ and $\ll$ is the bitwise matrix multiplication and bit-shift operation. The $\boxtimes$ can also be simulated as the common full-precision multiplication as the $\otimes$ during training.

We show the calculation process of the proposed $\boxtimes$ operation in Figure 7, which also concludes the process of $\otimes$.

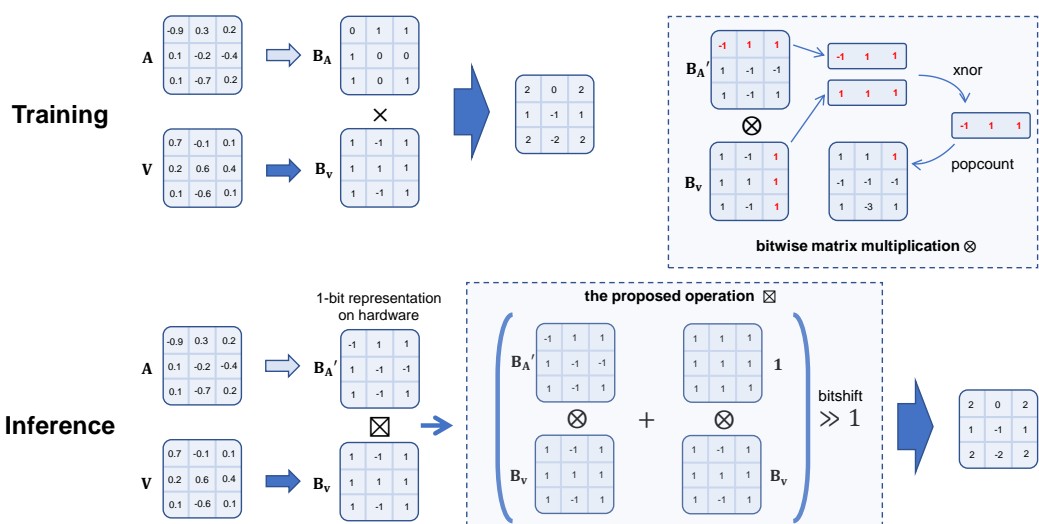

Figure 7: The calculation process of proposed ⊠ and ⊗ in Bi-Attention

### A.5 PROOF AND DISCUSSION OF THE THEOREM 3

**Theorem 4.** *Given the variables $X$ and $X_T$ follow $\mathcal{N}(0, \sigma_1), \mathcal{N}(0, \sigma_2)$ respectively, the proportion of optimization direction error is defined as $p_{error\ Q\text{-}bit} = p(\text{sign}(X - X_T) \neq \text{sign}(\text{quantize}_Q(X) - X_T))$, where $\text{quantize}_Q$ denotes the Q-bit symmetric quantization. As $Q$ reduces from 8 to 1, $p_{error\ Q\text{-}bit}$ becomes larger.*

*Proof.* Given the random variables $X \sim \mathcal{N}(0, \sigma_1)$ and $X_T \sim \mathcal{N}(0, \sigma_2)$, the $Q$-bit symmetric quantization function $\text{quantize}_Q$ is expressed as (take $X$ as an example)

$$\text{quantize}_Q(X) = \begin{cases} -L, & \text{if } x < -L, \\ \lfloor \frac{(2^Q-1)X}{2L} + 0.5 \rfloor \frac{2L}{2^Q-1}, & \text{if } -L \leq X \leq L, \\ L, & \text{if } x > L, \end{cases} \quad (61)$$

where the $\lfloor \cdot \rfloor$ denotes the round down function, and the range $[-L, L]$ is divided into $2^Q - 1$ inter. The optimization direction error occurs when $\text{sign}(X - \hat{X}) = \text{sign}(\text{quantize}_Q(X) - X_T)$, i.e., $X > X_T$ and $\text{quantize}_Q(X) < X_T$ **or** $X < X_T$ and $\text{quantize}_Q(X) > X_T$.

(1) When $-L < X < L$, $\lfloor \frac{(2^Q-1)X}{2L} \rfloor \frac{2L}{2^Q-1} < \text{quantize}_Q(X) < \lfloor \frac{(2^Q-1)X}{2L} + 1 \rfloor \frac{2L}{2^Q-1}$.

a) If $X_T < \lfloor \frac{(2^Q-1)X}{2L} \rfloor \frac{2L}{2^Q-1}$, since $\lfloor \frac{(2^Q-1)X}{2L} \rfloor \frac{2L}{2^Q-1} < X$, we have $X_T < X$. And since $\lfloor \frac{(2^Q-1)X}{2L} \rfloor \frac{2L}{2^Q-1} < \text{quantize}_Q(X)$, $X_T < \text{quantize}_Q(X)$. Thus, the optimization direction is always right in this case.

b) If $X_T > \lfloor \frac{(2^Q-1)X}{2L} + 1 \rfloor \frac{2L}{2^Q-1}$, since $\lfloor \frac{(2^Q-1)X}{2L} + 1 \rfloor \frac{2L}{2^Q-1} > X$, we have $X_T > X$. And since $\lfloor \frac{(2^Q-1)X}{2L} \rfloor \frac{2L}{2^Q-1} > \text{quantize}_Q(X)$, $X_T > \text{quantize}_Q(X)$. Thus, the optimization direction is always right in this case.

c) If $\lfloor \frac{(2^Q-1)X}{2L} \rfloor \frac{2L}{2^Q-1} \leq X_T \leq \lfloor \frac{(2^Q-1)X}{2L} + 1 \rfloor \frac{2L}{2^Q-1}$, first, the probability of $X > X_T$ and $\text{quantize}_Q(X) < X_T$ can be calculated as:

Table 4: Simulation of error proportion under the $Q$-bit

| Bits (Q) | 1 | 2 | 3 | 4 | 5 | 6 | 7 | 8 |
|---|---|---|---|---|---|---|---|---|
| **Proportion (%)** | 14.36% | 6.42% | 4.35% | 3.30% | 2.76% | 2.56% | 2.51% | 2.49% |

$$p_{\text{error1 } Q} = \int_{-L}^{L} \int_{\lfloor \frac{(2^Q-1)X}{2L} + 0.5 \rfloor \frac{2L}{2^Q-1}}^{X} f_{X,X_T}(X, X_T) \tag{62}$$

$$\left[ \lfloor \frac{(2^Q - 1)X}{2L} + 0.5 \rfloor < X \right] dX_T dX, \tag{63}$$

where $f_{X,X_T}(\cdot, \cdot)$ is the probability density function of the joint probability distribution for $\{X, X_T\}$, and $[\cdot]$ denotes the $Iverson\ bracket$ as defined in Eq. (26).

Then we get the probability of $X < X_T\ and$ $\text{quantize}_Q(X) > X_T$ as

$$p_{\text{error2 } Q} = \int_{-L}^{L} \int_{X}^{\lfloor \frac{(2^Q-1)X}{2L} + 0.5 \rfloor \frac{2L}{2^Q-1}} f_{X,X_T}(X, X_T) \tag{64}$$

$$\left[ \lfloor \frac{(2^Q - 1)X}{2L} + 0.5 \rfloor > X \right] dX_T dX. \tag{65}$$

Since $f_{X,X_T}(X, X_T) \geq 0$ is constant established, $p_{\text{error1 } Q}$ and $p_{\text{error2 } Q}$ increases as $Q$ becomes smaller.

(2) When $X > L$, $\text{quantize}_Q(X) = L$. $X_T > X > L = \text{quantize}_Q(X)$ is constant established when $X_T > X$, and when $X_T < X$, the probability of $X_T > \text{quantize}_Q(X) = L$ is also constant based on the given distribution of $X_T$. Thus, the optimization direction is always right in this case.

(3) When $X < -L$, $\text{quantize}_Q(X) = -L$. $X_T < X < -L = \text{quantize}_Q(X)$ is constant established when $X_T < X$, and when $X_T > X$, the probability of $X_T > \text{quantize}_Q(X) = -L$ is also constant based on the given distribution of $X_T$. Thus, the optimization direction is always right in this case.

$\square$

Although we have obtained the correlation between quantization bit-width and error of direction under symmetric quantization, it is difficult to directly give an analytical representation of the error proportion of Gaussian distribution input under Q-bit quantization. Therefore, we use the Monte Carlo algorithm to simulate the probability of directional error caused by $Q$-bit by the error proportion of the pre-quantized data $\mathbf{X} \in \mathbb{R}^{10000}$, where each element in $\mathbf{X}$ is sampled from the standard normal distribution. When the quantization range is $[-1, 1]$, the results are shown in Table 4. This result experimentally proved our theorem, and shows that the probability of direction mismatch increases rapidly in 1-bit quantization.

A.6   PROOF AND DISCUSSION OF THE EQUIVALENCE

Considering the order-preserving characteristic of both $\text{sign}$ and $\text{softmax}$, only the largest $n$ elements (the value of $n$ is related to the variable $k$ for specific matrix $\mathbf{A}$ in Theorem 1) are binarized to 1 while others are binarized to $-1$. Therefore, when $\text{sign}(\text{softmax}(\mathbf{A}) - \tau)$ is optimized to have the maximized information entropy, there exists a corresponding threshold $\phi(\tau, \mathbf{A})$ that maximizes the information entropy of $\text{sign}(\mathbf{A} - \phi(\tau, \mathbf{A}))$. In addition, the conclusion of Theorem 2 that $\mathbf{A} \sim \mathcal{N}(0, D)$ suggests that a fixed threshold $\phi(\tau, \mathbf{A}) = 0$ maximizes the information entropy of the binarized attention weight.

**Proposition 2.** *When the elements in tensor $\mathbf{A}$ are assumed as independent and identically distributed, there exists a $\phi(\tau, \mathbf{A}))$ making the entropy maximization of $\mathrm{sign}(\mathbf{A} - \phi(\tau, \mathbf{A}))$ be equivalent to that of $\mathrm{sign}(\mathrm{softmax}(\mathbf{A}) - \tau)$.*

*Proof.* As shown in Eq. (8), the information entropy of $\hat{\mathbf{B}}_{\mathbf{w}}^A = \mathrm{sign}(\mathrm{softmax}(\mathbf{A}) - \tau)$ can be expressed as:

$$\mathcal{H}(\hat{\mathbf{B}}_{\mathbf{w}}^A) = -\left(\int_\tau^\infty f(a^s)da^s\right)\log\left(\int_\tau^\infty f(a^s)da^s\right) - \left(\int_{-\infty}^\tau f(a^s)da^s\right)\log\left(\int_{-\infty}^\tau f(a^s)da^s\right),$$
$$(66)$$

where $a^s = \mathrm{softmax}_A(a)$ and $a \in \mathbf{A}$. Since $\mathrm{softmax}$ function is strictly order-preserving, there exists a $\phi(\tau, \mathbf{A})$ makes $\forall a > \tau, a^s > \phi(\tau, \mathbf{A})$, and it thereby makes following equations established:

$$\int_\tau^\infty f(a^s)da^s = \int_{\phi(\tau,A)}^\infty g(a)da, \quad \int_{-\infty}^\tau f(a^s)da^s = \int_\infty^{\phi(\tau,A)} g(a)da, \qquad (67)$$

where $f$ and $g$ are the probability density functions of variable $a^s$ and $a$. And the information entropy $\mathcal{H}(\hat{\mathbf{B}}_{\mathbf{w}}^A) = \mathcal{H}(\hat{\mathbf{B}}_{\mathbf{A}})$, where $\mathbf{B}_{\mathbf{A}} = \mathrm{sign}(\mathbf{A} - \phi(\tau, \mathbf{A}))$. Therefore, when $\tau$ is optimized to maximize information, the information of $\mathbf{B}_{\mathbf{A}}$ is also maximized under the threshold $\phi(\tau, \mathbf{A})$.

□

## B  Experimental Setup

### B.1  Dataset and Metrics

We evaluate our method on the General Language Understanding Evaluation (Wang et al., 2018a) (GLUE) benchmark which consists of nine basic language tasks. And we use the standard metrics for each GLUE task to measure the advantage of our method. We use Spearman Correlation for STS-B, Mathews Correlation Coefficient for CoLA and classification accuracy for the rest tasks. As for MNLI task, we report the accuracy on both in-domain evaluation MNLI-match (MNLI-m) and cross-domain evaluation MNLI-mismatch (MNLI-mm). But we exclude WNLI task as previous studies do for its relatively small data volume and unstable behavior. We also theoretically calculate the FLOPs at inference and also report the model size to give a comprehensive comparison on speed and storage.

We also provide a coarse estimation of model size and floating-point operations (FLOPs) at inference follow (Bai et al., 2020; Zhou et al., 2016; Liu et al., 2018). The matrix multiplication between an $m$-bit number and an $n$-bit number requires $mn/64$ FLOPs for a CPU with instruction size of 64-bit. Also, every 64 weight and embedding parameters are packed and stored together with a single instruction, so the model is significantly compact.

### B.2  Implementation

**Backbone**. We follow (Bai et al., 2020) to take full-precision well-trained DynaBERT (Hou et al., 2020) as the teacher model to self-supervise the training of the binarized DynaBERT. The number of transformer layers is 12, the hidden state size is 768 and the number of heads in MHA is 12, which is a typical setting of BERT-base. Unlike (Bai et al., 2020), we do not pre-train an intermediate model as the initialization, but directly binarize the model from the full-precision one. We use naive sign function to binarize parameters in the forward propagation and STE in the back-propagating, and also use MSE loss to distill the teacher model as described in Sec. 2.2 as the baseline binarized BERT model.

**Binarized Layers**. We follow the previous work to binarize the word embedding layer, MHA and FFN in transformer layers, but leave full-precision classifier, position embedding layer, and token

type embedding layer (Bai et al., 2020). That is because a common practice for BERT is to trivially cast the regression tasks to multiclass classification tasks, such as STS-B with 5 classes and MNLI with 3 classes. Therefore, the answers for above two tasks cannot be encoded by just 1-bit.

**Settings**. We use the Adam as our optimizer, and adopt data augmentation on GLUE tasks except MNLI and QQP for the little benefit but it is time-consuming. It is noteworthy that we take more training epochs for every quantization method on each tasks to have a sufficient training, which is 50 for CoLA, 20 for MRPC, STS-B and RTE, 10 for SST-2 and QNLI, 5 for MNLI and QQP.

**Other Quantized BERTs**. We implement TernaryBERT under 2-2-1 and 2-2-2 settings, and BinaryBERT under 1-1-1 and 1-1-2 settings for the compraisonal experiments.

For BinaryBERT (Bai et al., 2020), except 4-bit and 8-bit activation settings mentioned in their paper, the official code (`https://github.com/huawei-noah/Pretrained-Language-Model/blob/1dbaf58f17b8eea873d76aa388a6b0534b9ccdec/BinaryBERT/`) provides the implementations for 1-bit or 2-bit activation settings, which are Binary-Weight-Network (BWN) and Ternary-Weight-Network (TWN), respectively, and we completely follow the released code.

Specifically, we apply Binary-Weight-Network (BWN) and Ternary-Weight-Network (TWN) methods to activation under 1-bit and 2-bit input settings (1-1-1 and 1-1-2), respectively. The formulations and implementations are presented in (Bai et al., 2020) and official codes, respectively:

BWN:

$$\hat{a}_i^b = \mathcal{Q}\left(a_i^b\right) = \beta \cdot \text{sign}\left(a_i^b\right), \quad \beta = \frac{1}{n}\left\|\mathbf{a}^b\right\|_1, \tag{68}$$

TWN:

$$\hat{a}_i^t = \mathcal{Q}\left(a_i^t\right) = \left\{ \begin{array}{cc} \beta \cdot \text{sign}\left(a_i^t\right) & |a_i^t| \geq \Delta \\ 0 & |a_i^t| < \Delta \end{array} \right., \tag{69}$$

where $\text{sign}(\cdot)$ is the sign function, $\Delta = \frac{0.7}{n}\left\|\mathbf{w}^t\right\|_1$ and $\alpha = \frac{1}{|I|}\sum_{i \in \mathcal{I}}|w_i^t|$ with $\mathcal{I} = \{i \mid \hat{w}_i^t \neq 0\}$.

For TernaryBERT, since there is no recommended setting under 1-bit and 2-bit input in its official code, we use the same implementation as BinaryBERT to quantize activation (BWN for 1-bit and TWN for 2-bit) for fair comparison.

In addition, compared with TernaryBERT and BinaryBERT, under the 1-bit input setting, our BiBERT applies a simple sign function to binarize activation, while the former two apply real-time calculated scaling factors. Therefore, under the same 1-1-1 setting, our BiBERT achieved better results with a smaller amount of computation.

## C    ADDITION EXPERIMENTS

### C.1    COMPARISON AND DISCUSSION OF THE BASELINE

#### C.1.1    REASONS OF BUILDING THE ORIGINAL BASELINE

When we built the fully binarized BERT baseline, we considered the following reasons and chose the original baseline settings shown in the paper:

(1) Additional inference computation should be avoided to the greatest extent compared with the BERTs obtained by direct full binarization. We discard the scaling factor for activation in the bilinear unit since it requires real-time updating during inference and increases floating-point operations. In the attention structure, we directly binarize the activation without re-scaling or mean-shifting.

(2) Since there is no previous work to fully binarize BERT, the existing representative binarization techniques are chosen to build the baseline. The binarization function with a fixed 0 threshold is applied to the original definition of the binarized neural network (Rastegari et al., 2016) and is used by default in most binarization works (Qin et al., 2020; Liu et al., 2018), we thereby initially apply this function when building the fully binarized BERT baseline. Besides, it also helps to more objectively evaluate the benefits of maximizing information entropy in the attention structure.

Table 5: Comparison of strengthened baselines and methods without data augmentation.

| Solution | Quant | #Bits | $\Delta$FLOPs$_{(G)}$ | MNLI$_{\text{-m/mm}}$ | QQP | QNLI | SST-2 | CoLA | STS-B | MRPC | RTE | Avg. |
|---|---|---|---|---|---|---|---|---|---|---|---|---|
| Original | Baseline | 1-1-1 | 0 | 45.8/47.0 | 73.2 | 66.4 | 77.6 | 11.7 | 7.6 | 70.2 | 54.1 | 50.4 |
| | BinaryBERT | 1-1-1 | 0 | 35.6/35.3 | 63.1 | 51.5 | 53.2 | 0 | 6.1 | 68.3 | 52.7 | 40.6 |
| Solution 1 | Baseline$_{\text{asym}}$ | 1-1-1 | 0.074 (15%) | 45.1/46.3 | 72.9 | 64.3 | 72.8 | 4.6 | 9.8 | 68.3 | 53.1 | 48.6 |
| Solution 2 | Baseline$_{\mu}$ | 1-1-1 | 0.076 (16%) | 48.2/49.5 | 73.8 | 68.7 | 81.9 | 16.9 | 11.5 | 70.0 | 54.9 | 52.8 |
| Solution 3 | Baseline$_{50\%}$ | 1-1-1 | 0.076 (16%) | 47.7/49.1 | 74.1 | 67.9 | 80.0 | 14.0 | 11.5 | 69.8 | 54.5 | 52.1 |
| | BinaryBERT$_{50\%}$ | 1-1-1 | 0.076 (16%) | 39.2/40.0 | 66.7 | 59.5 | 54.1 | 4.3 | 6.8 | 68.3 | 53.4 | 43.5 |
| Ours | **BiBERT** | **1-1-1** | **0** | **67.3/68.6** | **84.8** | **72.6** | **88.7** | **25.4** | **33.6** | **72.5** | **57.4** | **63.5** |

Therefore, the fully binarized baseline under the current BERT architecture applies the representative techniques and almost without additional floating-point computation.

### C.1.2 SOLUTIONS OF ATTENTION WEIGHT

Causing by the application of $\text{softmax}$, the attention possibilities in the attention structure follow the probability distribution and (attention weight) are all one after binarization. In fact, we were aware of the problem when building the fully binarized baseline, and built two solutions by existing quantization techniques.

The first solution tried is to degenerate the asymmetric quantization function (usually applied to 2-8 bit quantization (Bai et al., 2020; Zhang et al., 2020)) to the 1-bit case, since the method is originally designed to deal with the imbalance of value distribution in quantization:

**Solution 1** (Baseline$_{\text{aysm}}$):

$$Q(x) = \begin{cases} \max(\mathbf{x}), & x \geq 0.5(\max(\mathbf{x}) + \min(\mathbf{x})), \\ \min(\mathbf{x}), & \text{otherwise}, \end{cases} \quad x \in \mathbf{x}. \tag{70}$$

The other solution is to simply use the mean of the elements as the threshold (Rastegari et al., 2016; Qin et al., 2020):

**Solution 2** (Baseline$_{\mu}$):

$$Q(\mathbf{x}) = \text{bool}(\mathbf{x} - \tau), \quad \tau = \mu(\mathbf{x}) \tag{71}$$

where $\mu(\cdot)$ denotes the mean value.

We also thank the reviewers for providing the Solution 3, which is to use the threshold limiting the zero attention weight to a certain percentage (such as 50%):

**Solution 3** (Baseline$_{50\%}$):

$$Q(\mathbf{x}) = \text{bool}(\mathbf{x} - \tau), \quad \tau = Q_{50\%}(\mathbf{x}) \tag{72}$$

where $Q_{50\%}(\cdot)$ denotes the quantile of 50% percentage (median).

The above three solutions are all able to alleviate the problem of fixed all one attention weight. We present the results of these solutions in Table A1.1 and further discuss them in detail.

**(1) Discussion of Solution 1**

As shown in Eq. (70), since the dynamically threshold $(0.5(\max(\mathbf{x}) + \min(\mathbf{x})))$ is applied in this quantizer, the elements of attention weight are quantized to $\max(\mathbf{x})$ and $\min(\mathbf{x})$ instead of a same value. However, the asymmetrical binarized values obtained by this quantizer make the binarized BERT hard to apply bitwise operations and lose the efficiency advantage brought by binarization (0.074G additional FLOPs), so that this practice is not used in existing binarization works. And as the results show, the accuracy of the binarized BERT is also worse than the existing baseline (Baseline 50.4% vs. Baseline$_{\text{asym}}$ 48.6% on average, Row 2 and 4 in Table 5).

**(2) Discussion of Solution 2**

Table 6: Ablation results with 10%~90% zero attention weight on STS-B.

| Percentage | 10% | 30% | 50% | 70% | 90% |
|---|---|---|---|---|---|
| **Entropy** | 0.47 | 0.88 | **1.00** | 0.88 | 0.47 |
| **Accuracy (%)** | 8.5 | 8.6 | **11.5** | 10.8 | 10.0 |

As the effect of this solution on the weight parameter (Rastegari et al., 2016; Qin et al., 2020), it can ensure diversity of binarized attention weight elements instead of all 1 ($\min(\mathbf{x}) \leq \tau \leq \max(\mathbf{x})$). But the premise of this practice is the Gaussian (symmetric) distribution of floating-point parameters before binarization (Qin et al., 2020), while the attention score follows an asymmetric probability distribution, causing the values of binarized activation to imbalanced. Therefore, the improvement of this practice in terms of accuracy is also limited (as shown in Row 2 and 5 in Table 5), and this solution also increases 0.076G computational FLOPs compared to the original baseline.

**(3) Discussion of Solution 3**

The fixed 50% percentage of zero binarized attention weights suggested by the reviewer are good baseline settings that help to further clarify the entropy maximization motivation of the Bi-Attention in BiBERT. As shown in Table 5 (Row 6 and 7), the results of baseline and BinaryBERT under this setting are significantly improved by 1.7% and 2.9% on average compare the original results (Row 2 and 3), respectively. And from the perspective of information entropy, a 50% percentage of zero attention weights can also ensure maximum information. We present the detailed ablation results (10%-90% zero attention weight) on STS-B in Table 6, which show the model maximizing information entropy achieves the best results.

Though forcing a certain ratio of zero attention weights improves the baseline, there exists shortcomings for fully binarized BERTs. First, the calculation of quantile thresholds relies on real-time computation or sorting, which increases computation of the fully binarized BERT in inference (about 0.076G (16%) additional FLOPs). Moreover, the practice of 50% quantile threshold should be regarded as the solution to a more stringent optimization problem (rather than the optimization problem in Eq. (9) of the paper) since it further constrains and maximizes the entropy of each tensor of binarized attention weight instead of optimizing the overall distribution. Thus, the quantile threshold is more restrictive for the binarized attention weights and limits their representation capability, causing the results of obtained fully binarized BERTs lower than those of models applying Bi-Attention.

### C.1.3    CONCLUSION

Although the above three solutions improve accuracy to a certain extent, they only bring limited improvements while destroying the advantages of the fully binarized network on computational efficiency. As analyzed in our paper, the existing attention structure is not suitable for directly fully binarizing BERT, which is also the reason we specially designed the Bi-Attention structure for the fully binarized BERT. However, considering the reason that the fixed all one attention weight causes itself and related distillation to fail, and also to more comprehensively compare with our method, we present the results of these solutions (as strengthened baselines).

### C.2    VARIANTS OF BINARIZED ATTENTION MECHANISM

In binarization literature, function $\mathrm{sign}$ is widely used to quantize a real number $x \in \mathbb{R}$ into $y \in \{-1, 1\}$. However, directly applying $\mathrm{sign}$ on attention weight is against the constraint that attention weight should be in $[0, 1]$. Inspired by this fact, we define function $\mathrm{bool}$ to map $x$ into $\{0, 1\}$. We empirically evaluate these two binarization methods on several GLUE tasks in Table 7.

Table 7 shows the performance of four different binarized attention mechanisms. All the experiments use the DMD method and do not use data augmentation. From these results, we can conclude that the traditional binarization method $\mathrm{sign}$ is not suitable to binarize attention weight since it can not selectively neglect the unrelated parts. This operation not only obstacles the optimization process but also degrades the representation ability of the whole attention mechanism. It strongly shows the

Table 7: Comparison of Variants of Binarized Attention Mechanism

| | Maximizing Entropy | Binarization Method | SST-2 | RTE | CoLA |
|---|---|---|---|---|---|
| 1 | ✗ | sign | 78.9 | 52.7 | 6.9 |
| 2 | ✗ | bool | 77.6 | 54.1 | 11.7 |
| 3 | ✓ | sign | 80.7 | 52.7 | 7.3 |
| 4 | ✓ | bool | 88.7 | 57.4 | 25.4 |

necessity of using a binarization operator that can focus selectively on parts of the input and ignore the others.

## D  VISUALIZATIONS

In Figure 8, we provide a coarse but bird-view of attention patterns across layers and heads in different model for same inputs on SST-2 task, including the full-precision, fully binarized BERT baseline and our BiBERT. As can be seen in the figure, full-precision model always elicits certain tendency towards different contents, and each head captures a unique pattern with bias. However, the attention perception is totally lost when fully binarized the BERT model with baseline methods. The attention weight is identical regardless of inputs, and exactly degrades to a fully connection.

As for our BiBERT, it delivers various attention patterns among layers and heads. We concede that some patterns seem to be completely different from the full-precision ones, some are even blank. But we venture to attributes the phenomenon to the special design of our Bi-Attention structure. We ensure the overall probability of attention weights to have a information entropy maximized distribution, but the function and quality of each head is special after an adaptive training. It results in a selective attention according to the inputs, therefore we revive the perception of the model, and make it be able to elicit more diverse activation patterns.

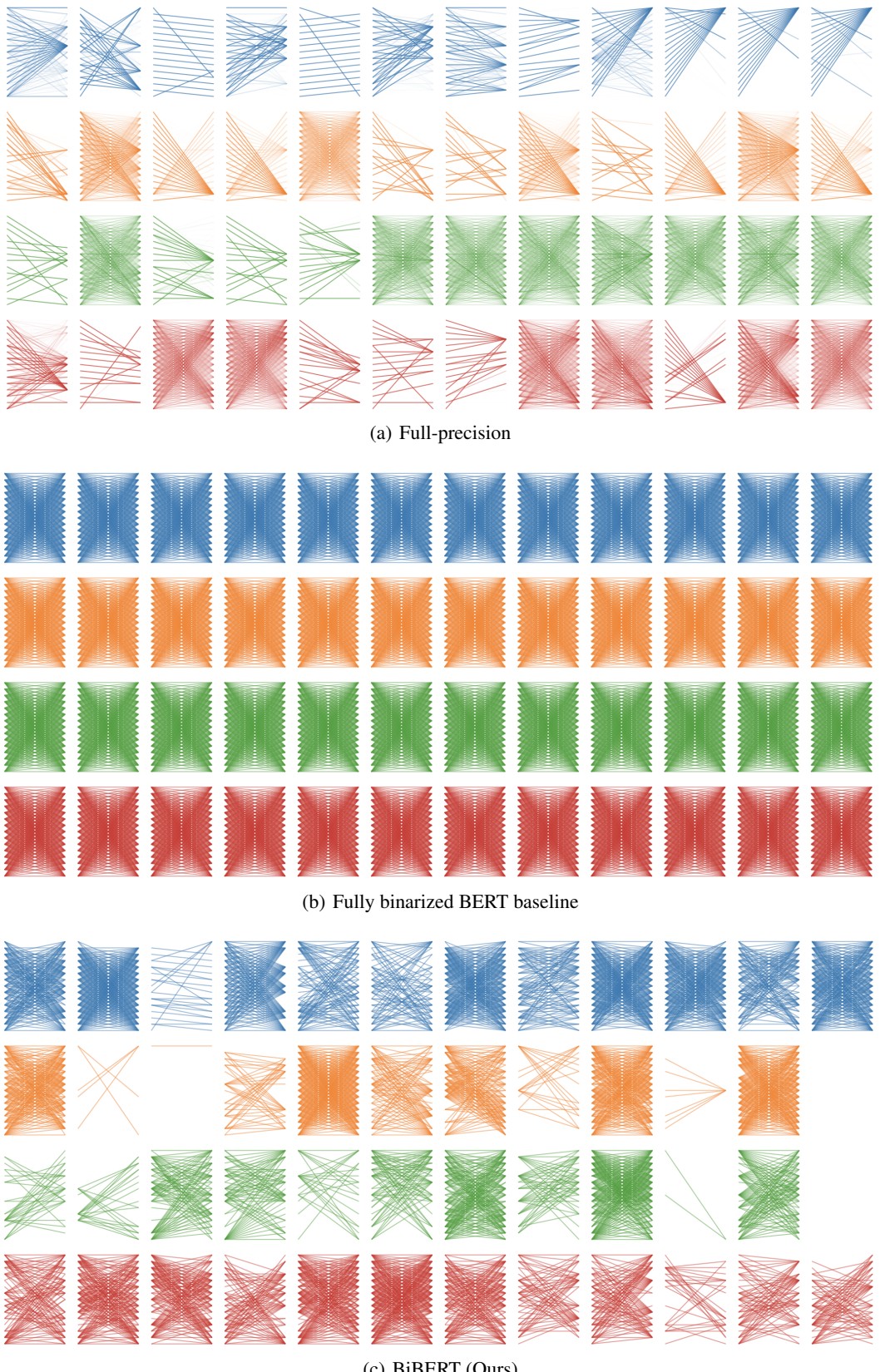

(a) Full-precision

(b) Fully binarized BERT baseline

(c) BiBERT (Ours)

Figure 8: Attention pattern for full-precision model, fully binarized BERT baseline and our BiBERT on SST-2 task. Includes all heads in layers 0-3. The visualization tools is adapted from (Vig, 2019)

