# OpenReview forum: "BiBERT: Accurate Fully Binarized BERT"
_ICLR.cc/2022/Conference — ICLR 2022 Poster_

### Official Review · Reviewer_TtMP · 2021-11-02

**Correctness:** 3
**Technical Novelty And Significance:** 3
**Empirical Novelty And Significance:** 2
**Recommendation:** 6
**Confidence:** 4

**Main Review:**

Overall, the paper is well-motivated and easy to follow. To the best of my knowledge, this is the first work on full quantization (weights-embedding-activation) of the BERT model to 1-bit (denoted as 1-1-1), considering that previous work such as BinaryBERT only achieves 1-1-4 quantization. Binarizing all the activations without losing much performance is challenging. By making detailed analysis of the performance variances in binarizing intermediate outputs at each step and the different distillation losses, the authors find the major problems that hurt the model performance most. Two new methods are proposed to avoid or alleviate the problem in training the quantized model, which are theoretical sound and shows substantial improvements to the accuracies on downstream tasks. The proposed BiBERT model also surpasses the previous BERT quantization model on the setting of 1-1-1 quantization.

The weaknesses of the BiBERT model are as follows:
1) The performance of BiBERT on GLUE is still far behind the full precision model (67.0 vs 82.3). This large performance gap may hinder application of the BiBERT model in many real scenarios (e.g., CoLA, STS-B tasks). While the BinaryBERT with 1-1-4 quantization, which has similar model size and a little bit higher computation as the BiBERT, can achieve GLUE score of 81.9 with almost neglectable performance loss. This raises the question why full binarization of the BERT model is necessary or preferred. One possible way to justify this is to enlarge the network size while keeping it binary and see if the performance can consistently improve.
2) In the experiments, the authors only report BinaryBERT (1-1-2 and 1-1-1) and TernaryBERT (2-2-1 and 2-2-2) which are not present in their original papers. The author should clarify in the paper how the results are obtained. Are the results from reimplementation of their algorithms or based on some open-source code/models? Are the reimplementation correct? On CoLA task, BinaryBERT and TenaryBERT simply seem to diverge, and the baseline quantization method without Bi-Attention and DMD also outperform BinaryBERT and TenaryBERT.
3) The author report Q2BERT and Q-BERT on the 2-2-8 setting, while the TernaryBERT have much higher performance (72.9) on it. Why not report these results?
Overall, I think the BiBERT is still not applicable to some NLU tasks given severe performance drop. Some of the experimental results are not convincing (or biased) regarding the baseline of BinaryBERT and TenaryBERT.


**Summary Of The Paper:**

This paper addresses the problem of fully binarizing BERT model including the network weights, embeddings and activations. Through theoretical and empirical analysis, the authors find: 1) the direct binarization of softmax-ed attention matrix is problematic; 2) distillation of the attention scores from the full-precision teacher can be harmful to the binarized model. Therefore, they propose two different methods to alleviate the issues. One is the Bi-Attention which directly quantizes the attention matrix and excludes softmax. The other is Direction-Matching Distillation which chooses to distill the covariance matrices of query/key/value rather than the attention matrix. Experimental results on GLUE benchmark show that BiBERT model outperforms previous quantized BERT models on the full binarization setting.
This main contribution of the paper is that it is the first work on full binarization of BERT models. With detailed analysis and observations, it proposed two effective methods to improve the training of the binarized model, and achieve good performance on some of the NLU tasks.

**Summary Of The Review:**

The paper tries to tackle a challenging problem of binarizing BERT and find effective ways to improve the performance. However, the model performances on some tasks are still far from good and comparison in the experiments are not convincing. Therefore, my current recommendation is weak reject unless the author can clarify the concerns.

---

> ### Author Response · Authors · 2021-11-15
> **Response to Reviewer TtMP (1/3)**
>
> We thank the reviewer for the constructive feedback and comments. We respond to the concerns below:
>
>
>
> **Q1:** The performance of BiBERT on GLUE is still far behind the full precision model (67.0 vs 82.3). This large performance gap may hinder application of the BiBERT model in many real scenarios (e.g., CoLA, STS-B tasks). While the BinaryBERT with 1-1-4 quantization, which has similar model size and a little bit higher computation as the BiBERT, can achieve GLUE score of 81.9 with almost neglectable performance loss. This raises the question why full binarization of the BERT model is necessary or preferred. One possible way to justify this is to enlarge the network size while keeping it binary and see if the performance can consistently improve.
>
> **A1:** We need to state that the fully binarized (1-1-1 bit-width) BERT has great computational efficiency advantages benefiting from the extremely efficient bitwise operations in the inference, while the BERT with binarized weight and 4 or 8-bit activation (such as 1-1-4/1-1-8 BinaryBERT [1]) conducts integer operations which is far less efficient. Fully binarized parameters bring extremely efficient XNOR-Bitcount bitwise operations [2] with ultra-low energy consumption (achieve $89\times$ energy efficiency) [3] and lower storage usage. The full binarization allows the network to be compressed and deployed on limited computing resources and ultra-low power devices, such as mobile phones and cameras, and also enables the applications that require real-time interaction and fast response to run steadily on edge devices. However, the extremely limited representation capabilities and the difficult discrete optimization make the accurate fully binarized networks a great challenge.
>
> BiBERT is a fully binarized network that applies 1-bit weight, activation, and embedding, and utilizes bitwise operations to achieve a $56.3\times$ saving in FLOPs compared to full-precision BERT. By contrast, BinaryBERT [1] keeps activations to 4-bit which suffers integer operations and $3.8\times$ FLOPs consumption compared to BiBERT. Therefore, BiBERT is significant since it first realizes the convergence of fully binarized BERT, which enjoys high computational FLOPs reduction and storage-saving ($56.3\times$ and $31.2\times$, respectively). We admit that the performance of BiBERT on GLUE is still behind the full precision model, but it far exceeds the fully binarized models obtained by existing BERT quantization methods. And just like the model binarization widely studied in other fields (e.g. image classification, object detection, point cloud processing), full binarization of NLU models is prospective, and the accuracy will constantly approach SOTA with deeper studies in the future.

---

> > ### Author Response · Authors · 2021-11-15
> > **Response to Reviewer TtMP (2/3)**
> >
> > **Q2 (1):** In the experiments, the authors only report BinaryBERT (1-1-2 and 1-1-1) and TernaryBERT (2-2-1 and 2-2-2) which are not present in their original papers. The author should clarify in the paper how the results are obtained. Are the results from reimplementation of their algorithms or based on some open-source code/models? Are the reimplementation correct?
> >
> > **A2 (1):** Thanks for pointing out that. We update more details about the implementations of BinaryBERT [1] and TernaryBERT [4] under 1-bit and 2-bit activation settings in the revision (Appendix B.2).
> >
> > BiBERT is the first work towards the fully binarized BERT with 1-bit weight, activation, and embedding, and enjoys extremely high computational efficiency and memory usage as we discussed in A1. To be fair, all the networks in the comparative experiments should be set to the same 1-1-1 bit-width to have a similar computation and storage level, otherwise, the accuracy comparison does not make sense. However, there is no existing BERT quantization method that binarizes activation, we thereby have to use the most closely related works BinaryBERT and TernaryBERT as the major competitors, which quantize weight and embedding to ultra-low bit-width (1-bit or 2-bit), and further reduce the activation to 1-bit or 2-bit to compare the accuracy rate under similar computational and storage efficiency. The quantization processes of activation follow official or fair implementation as much as possible.
> >
> > For BinaryBERT, though it is not mentioned in their paper, the official code provides an implementation for 1 or 2-bit inputs quantization, which is Binary-Weight-Network (BWN) and Ternary-Weight-Network (TWN) respectively. And we completely follow the released code. The implementations of BWN and TWN can be formulated as follows:
> >
> > BWN: $\mathcal{Q}(a_{i}^{b})=\beta \cdot \operatorname{sign}\left(a_{i}^{b}\right), \beta=\frac{1}{n}\left\|\mathbf{a}^{b}\right\|_{1},$
> >
> > TWN: $\mathcal{Q}\left(a_{i}^{t}\right)=\beta \cdot \operatorname{sign}\left(a_{i}^{t}\right), \left|a_{i}^{t}\right| \geq \Delta; 0,  \left|a_{i}^{t}\right|<\Delta.$
> >
> > Please refer to BinaryBERT official repo for implementation details: https://github.com/huawei-noah/Pretrained-Language-Model/blob/1dbaf58f17b8eea873d76aa388a6b0534b9ccdec/BinaryBERT/transformer/utils_quant.py#L425
> >
> > For TernaryBERT, since there is no recommended setting under 1-bit and 2-bit input in its official code, we use the same implementation as BinaryBERT to quantize activation (BWN for 1-bit and TWN for 2-bit) for fair comparison.
> >
> > In fact, existing work [1] has also pointed out that quantizing the activation is of great challenge and the ultra-low bit-width activation quantization on the existing method may lead to too bad results, which is exactly the problem our BiBERT devotes to solve.  Compared with TernaryBERT and BinaryBERT, under the 1-bit input setting, our BiBERT applies a simple sign function to binarize activation, while the former two apply real-time calculated scaling factors. Therefore, benefitting from the well-designed structure and distillation scheme, our BiBERT achieved better results with a smaller amount of computation.

---

> > > ### Author Response · Authors · 2021-11-15
> > > **Response to Reviewer TtMP (3/3)**
> > >
> > > **Q2 (2):** On CoLA task, BinaryBERT and TenaryBERT simply seem to diverge, and the baseline quantization method without Bi-Attention and DMD also outperform BinaryBERT and TenaryBERT.
> > >
> > > **A2 (2):** For the results of TernaryBERT (2-2-1 and 2-2-2) and BinaryBERT (1-1-2 and 2-2-1) CoLA and other diverged tasks (such as RTE), the severely limited representation ability and optimization difficulties caused by the ultra-low bit activation are the main reasons for the sharp drop in the accuracy, while the direct application of existing methods cannot solve these problems.
> > >
> > > Specifically, compared with 1-1-1 BinaryBERT, our fully binarized BERT baseline applies the bi-linear defined in Eq. (2) throughout the network, which significantly improves the binarized network by maximizing the information entropy of binarized weight and activation. Our further experiments show the effectiveness of bi-linear with zero-mean weight on some tasks that 1-1-1 BinaryBERT crashed (CoLA and RTE) in Table A2.1, which shows that it significantly improves the accuracy and allows the model to converge.
> > >
> > > **Table A2.1: Comparison results on CoLA and RTE**
> > >
> > > | Method                 | #Bit  | CoLA | RTE   |
> > > | ---------------------- | ----- | ---- | ----- |
> > > | BinaryBERT             | 1-1-1 | 0.0% | 52.7% |
> > > | BinaryBERT (bi-linear) | 1-1-1 | 8.3% | 54.2% |
> > >
> > > Moreover, as analyzed in our paper, though the information entropy maximized bi-linear is applied, fully binarizing BERTs under the existing architecture and training scheme (such as the 1-1-1 BinaryBERT and fully binarized BERT baseline) leads to severe information degradation and optimization of mismatch directions, which results in a sharp drop in the accuracy and even divergence in certain tasks. BiBERT further significantly relieves the above issues and improves accuracy, which makes it a pioneer work of the accurate fully binarized BERT.
> > >
> > >
> > >
> > > **Q3:** The author report Q2BERT and Q-BERT on the 2-2-8 setting, while the TernaryBERT have much higher performance (72.9) on it. Why not report these results?
> > >
> > > **A3:** As mentioned in A2(1), all BERTs for comparison should be set to similar bit-widths for fairness, and we use the most closely related works BinaryBERT and TernaryBERT as the major competitors in our paper. The 2-2-8 TernaryBERT truly performs better in terms of accuracy, but it suffers about $30\times$ FLOPs computation and $2\times$ and storage usage compared with the fully binarized BERT, which leads to an unfair comparison. Though the 1-1-1 BiBERT even surpasses the 2-2-8 Q-BERT and QBERT due to its well-designed architecture and training scheme (we directly quote the results in [4]), it does not mean that BiBERT is able to exceed all methods under 2-2-8 bit-width setting.
> > >
> > >
> > >
> > > **Reference**
> > >
> > > [1] Bai, Haoli, et al. "BinaryBERT: Pushing the Limit of BERT Quantization." ACL. 2021.
> > >
> > > [2] Rastegari, Mohammad, et al. "Xnor-net: Imagenet classification using binary convolutional neural networks." ECCV. 2016.
> > >
> > > [3] Chen, Gang, et al. "PhoneBit: efficient gpu-accelerated binary neural network inference engine for mobile phones." DATE. 2020.
> > >
> > > [4] Zhang, Wei, et al. "TernaryBERT: Distillation-aware Ultra-low Bit BERT." EMNLP. 2020.

---

> > > > ### Comment · Reviewer_TtMP · 2021-11-30
> > > > **Response to Authors**
> > > >
> > > > I thank the authors for the detailed feedback, although they havn't fully addressed my concern regarding the large performance drop of the BiBERT. And again, it is important to make the comparison fair and not biased toward your own method. Overall, I think the work has its merits.

---

### Official Review · Reviewer_84Py · 2021-11-02

**Correctness:** 3
**Technical Novelty And Significance:** 3
**Empirical Novelty And Significance:** 2
**Recommendation:** 6
**Confidence:** 3

**Main Review:**

Strengths:
1. The paper is well constructed and easy to go through the idea; encompasses rigorous theoretical justification of the approaches.
2. This paper addresses the crucial problems in fully binarized BERT and introduces BiBERT which is the very first attempt to implement an accurate fully binarized BERT.
3. In this framework, they incorporate a Bi-Attention approach which maximizes the information entropy to mitigate the information degradation. And a Direction Matching Distillation (DMD) mechanism which resolves the optimization direction mismatch.
4. BiBERT also impressively obtain savings on FLOPs and model size which is a great advantage for real-world edge devices.
5. It also contains an ablation study on the GLUE benchmark which validates the contribution of the individual components.

Weaknesses:
1. The proposed method didn’t properly address the performance drop due to the binarization of activation. Results on the GLUE benchmark showed a drastic performance drop in comparison to the related approaches and full precision BERT.
2. In sec 2.1, it is evident that a stochastic sign function is used to binarize the BERT, therefore, why the deterministic function is used in sec 3.2.1 is not clear.
3. In table 1,2, 3, we see that data augmentation method has been utilized for experimental purposes. But it is not properly addressed/clarified in sec 4.1 and 4.2 why it is employed and what is the significance of this approach here.
4. Also, these two closely related works should be critically analyzed in the experiment section:
- Bai, H., Zhang, W., Hou, L., Shang, L., Jin, J., Jiang, X., ... & King, I. (2020). Binarybert: Pushing the limit of bert quantization. arXiv preprint arXiv:2012.15701.
- Zhang, W., Hou, L., Yin, Y., Shang, L., Chen, X., Jiang, X., & Liu, Q. (2020). Ternarybert: Distillation-aware ultra-low bit bert. arXiv preprint arXiv:2009.12812.
5. Figure 2, 3, 4 are not mentioned in appropriate sections to help understand the analysis.

Some minor issues and typos:
1. There is no full stop(.) at the end of each caption of the figures and tables.
2. At the end of the 2nd paragraph of the Introduction: “So far, previous studies have pushed down the weight and embedding to be binarized, but none of them have ever achieved to binarize BERT with 1-bit activation accurately.” -> please refer to the specific previous studies.
3. Section 2, first paragraph: BRET -> BERT
4. Section 2.2: which can be unobstructed applied -> which can be unobstructedly applied
5. Appendix B.2: There is no citation for DynaBERT.


**Summary Of The Paper:**

This paper introduces an accurate fully binarized (i.e., 1-bit weight, embedding, and activation)  BERT, BiBERT, as a robust model compression method. It saves 56.3× and 31.2× on FLOPs and model size for real-world devices. Moreover, it addresses the substantial performance issues in the straightforward full binarization method. To tackle the performance drop issues, it proposes Bi-Attention and Direction Matching Distillation (DMD) mechanism.  The paper contains an evaluation of the proposed model on GLUE benchmark and a comparison with SOTA quantized BERT models.


**Summary Of The Review:**

This paper portrays the related performance issues of straightforward fully binarized BERT and introduces the idea of BiBERT as a solution. The paper has proper experimental proofs and analysis to justify the framework. It might be a positive inclusion towards an accurate fully binarized BERT architecture. The proposed method obtained impressive savings on FLOPs and model size. However, it seems that the related SOTA approaches are missing in the comparative performance analysis section, and I would really be looking to see this in a conference paper that meets ICLR's high standards.

---

> ### Author Response · Authors · 2021-11-15
> **Response to Reviewer 84Py (1/2)**
>
> We thank for your professional and insightful comments. Our response to your suggestion can be found below:
>
>
>
> **Q1:** The proposed method didn’t properly address the performance drop due to the binarization of activation. Results on the GLUE benchmark showed a drastic performance drop in comparison to the related approaches and full precision BERT.
>
> **A1:** Although BiBERT has extreme computational savings and achieves significant progress on most tasks, we admit that there is still a significant gap in accuracy between the fully-binarized and full-precision BERTs. In fact, full binarization presents a great potential for large-scale language models. And there is a huge demand for NLP models to run on limited computing resources and ultra-low power devices, such as mobile phones and cameras. Recently, fully binarized networks have been widely studied in many fields (e.g., image classification [1], data generation [2], object detection [3], point cloud processing [4]) for the extremely compact 1-bit parameters and efficient bitwise operations. These models achieve real-time interaction and fast response after quantization on source-limited devices. Compared to these successful practices, large pre-trained language models have greater demands for model quantization since the larger parameter sizes and more computational cost. Moreover, the fully binarized networks are also constantly approaching SOTA accuracy in these fields [4,5,6] with the proposing of well-designed architectures and training schemes.
>
> Our BiBERT, for the first time, presents a promising route towards the accurate fully binarized BERT for NLP. As analyzed in our paper, fully binarizing BERTs under the existing architecture and training scheme (such as the BinaryBERT and fully binarized BERT baseline) leads to severe information degradation and optimization of mismatch directions, which leads to a sharp drop in the accuracy and even divergence in certain tasks. As a pioneer work, BiBERT first realizes the convergence of fully binarized BERT by significantly relieving the above issues, while enjoying ultra-high computational FLOPs reduction and storage savings ($56.3\times$ and $31.2\times$, respectively), and also far exceeds the fully binarized models obtained by existing BERT quantization methods. We believe that the fully binarized BERT will develop in the future, and will continue to approach SOTA accuracy while maintaining ultra-high computing and storage efficiency.
>
>
>
> **Q2:** In sec 2.1, it is evident that a stochastic sign function is used to binarize the BERT, therefore, why the deterministic function is used in sec 3.2.1 is not clear.
>
> **A2:** We clarify the sign function used in our whole paper is deterministic as defined in Eq. (1) of Section 2.1.
>
> In the papers proposing STE [7] and some early binarized networks [8], the stochastic sign function was applied or mentioned. However, in recent and the most proposed binarized networks [1,2,3,4,5,6], the deterministic function is routinely adopted for the advantages of computational efficiency and hardware implementation. Therefore, we use a deterministic binarization function in the forward propagation and STE in the backward propagation.
>
>
>
> **Q3:** In table 1,2, 3, we see that data augmentation method has been utilized for experimental purposes. But it is not properly addressed/clarified in sec 4.1 and 4.2 why it is employed and what is the significance of this approach here.
>
> **A3:** Thank you for pointing it out. We further clarify the method and significance of data augmentation in the revision.
>
> We apply the data augmentation proposed in [9] as a strategy to increase the diversity of data available for the training models, which is orthogonal to model binarization and beneficial to all the BERT training methods. Previous studies like BinaryBERT [10] and TernaryBERT [11] also use the same data augmentation method as one of the default settings in the experiment. Thus, we evaluated BiBERT and other methods on augmented data for fair comparison. Moreover, for comprehensive comparison, we also include experiments without data augmentation. Experiments show that BiBERT performs better than other methods whether data augmentation is used or not.

---

> > ### Author Response · Authors · 2021-11-15
> > **Response to Reviewer 84Py (2/2)**
> >
> > **Q4:** Also, these two closely related works should be critically analyzed in the experiment section:
> >
> > - Bai, H., Zhang, W., Hou, L., Shang, L., Jin, J., Jiang, X., ... & King, I. (2020). Binarybert: Pushing the limit of bert quantization. arXiv preprint arXiv:2012.15701.
> > - Zhang, W., Hou, L., Yin, Y., Shang, L., Chen, X., Jiang, X., & Liu, Q. (2020). Ternarybert: Distillation-aware ultra-low bit bert. arXiv preprint arXiv:2009.12812.
> >
> > **A4:** Thanks for your constructive suggestions. We add more discussions and deeper analysis about these methods [10,11] in the experiment section.
> >
> > Our BiBERT surpasses existing methods on BERT$_\text{BASE}$ architecture by a wide margin in the average accuracy, and first achieves convergence under ultra-low bit activation on some tasks, such as CoLA, MRPC, and RTE. While under ultra-low bit activation, the accuracy of TernaryBERT and BinaryBERT decreases severely which also does not improve under the 2-2-2 setting (about $4\times$ FLOPs and $2\times$ storage usage increase). On some specific tasks like MRPC, higher bit-widths and consumption did not even make TernaryBERT and BinaryBERT converge. We also noticed that BiBERT lags behind Q-BERT (2-8-8) and TernaryBERT (2-2-2) on MNLI and STS-B (with DA) while surpassing them on other tasks, indicating the fully binarized BERT may have greater improvement potential in these tasks.
> >
> >
> >
> > **Q5:** Figure 2, 3, 4 are not mentioned in appropriate sections to help understand the analysis.
> >
> > **A5:** Thanks for pointing it out. We have rearranged the figure positions to better help the illustration.
> >
> >
> >
> > **Q6:** Some minor issues and typos.
> >
> > **A6:** Thanks for pointing it out. We have carefully revised these issues and refined our paper in the revision.
> >
> >
> >
> > **Reference**
> >
> > [1] Martinez, Brais, et al. "Training binary neural networks with real-to-binary convolutions." ICLR. 2019.
> >
> > [2] Bird, Thomas, Friso Kingma, and David Barber. "Reducing the Computational Cost of Deep Generative Models with Binary Neural Networks." ICLR. 2020.
> >
> > [3] Wang, Ziwei, et al. "BiDet: An efficient binarized object detector." CVPR. 2020.
> >
> > [4] Qin, Haotong, et al. "BiPointNet: Binary Neural Network for Point Clouds." ICLR. 2020.
> >
> > [5] Liu, Zechun, et al. "Reactnet: Towards precise binary neural network with generalized activation functions." ECCV. 2020.
> >
> > [6] Xu, Yixing, et al. "Learning Frequency Domain Approximation for Binary Neural Networks." NeurIPS. 2021.
> >
> > [7] Bengio, Yoshua, Nicholas Léonard, and Aaron Courville. "Estimating or propagating gradients through stochastic neurons for conditional computation." arXiv preprint arXiv:1308.3432 (2013).
> >
> > [8] Courbariaux, Matthieu, et al. "Binarized neural networks: Training deep neural networks with weights and activations constrained to+ 1 or-1." arXiv preprint arXiv:1602.02830 (2016).
> >
> > [9] Jiao, Xiaoqi, et al. "TinyBERT: Distilling BERT for Natural Language Understanding." EMNLP Findings. 2020.
> >
> > [10] Bai, Haoli, et al. "BinaryBERT: Pushing the Limit of BERT Quantization." ACL. 2021.
> >
> > [11] Zhang, Wei, et al. "TernaryBERT: Distillation-aware Ultra-low Bit BERT." EMNLP. 2020.

---

> > > ### Comment · Reviewer_84Py · 2021-11-18
> > > **Response to Authors**
> > >
> > > Thanks to the authors for the response. They clarified most of my questions. But the main concern remains the same i.e. comparison with SOTA approaches instead of setting them with BiBERT settings. I think this paper should address the comparison with the SOTA methods as a standard paper. The inclusion of the SOTA methods can help demonstrate the differences and highlight your novel contribution towards the binarization of activation. Otherwise, it seems that the paper is too biased to your findings.

---

> > > > ### Author Response · Authors · 2021-11-19
> > > > **Re: Response to Authors**
> > > >
> > > > Thanks for your positive and very helpful feedback! We fully agree with your suggestions and further add and compare the results of existing SOTA BERT quantization methods (1-1-4 BinaryBERT [1] and 2-2-8 TernaryBERT [2] which are reported in their original papers) in Table 2 and Table 3 of our new revision.
> > > >
> > > > As the results show, though the BERTs quantized by existing methods [1,2] almost maintain the accuracy under 4-bit or above bit-widths activation, they sharply crash when the activation is binarized/quantized to 1-bit and 2-bit. Moreover, even though BinaryBERT binarizes the weight and embedding to 1-bit for the first time, activation binarization can further bring up to $3.8\times$ of computational FLOPs saving and allows the model to apply extreme efficient bitwise operations instead of integer operations. The above facts make the fully binarized BERT (activation binarization of BERT) an approach with both significant attraction and challenge.
> > > >
> > > > As a pioneer work, BiBERT presents a promising route towards the accurate fully binarized BERT (with 1-bit weight, embedding, and activation) for the first time. The experiments show that our BiBERT outperforms existing quantized BERTs with 1-bit/2-bit activation by convincing margins. And though a significant gap still exists with the full-precision model, BiBERT presents the ultra-high $56.3\times$ and $31.2\times$ computational FLOPs and storage savings, respectively. Thus, this work confirms that full binarization is one of the most promising compression approaches for large-scale pre-trained BERT that significantly alleviates the deployment difficulties of BERTs in real-world hardware with limited computation and storage resources.
> > > >
> > > > [1] Bai, Haoli, et al. "BinaryBERT: Pushing the Limit of BERT Quantization." ACL. 2021.
> > > >
> > > > [2] Zhang, Wei, et al. "TernaryBERT: Distillation-aware Ultra-low Bit BERT." EMNLP. 2020.

---

### Official Review · Reviewer_tpUQ · 2021-11-02

**Correctness:** 3
**Technical Novelty And Significance:** 2
**Empirical Novelty And Significance:** 3
**Recommendation:** 5
**Confidence:** 4

**Main Review:**

The authors stated that the bi-linear quantization function in Equation (2) is motivated by Rastegati et al., 2016 and Qin et al., 2021.  However, (2) is quite different from the quantization methods in both papers. Specifically,
- in Rastegati et al., 2016, both the weights and activations are not zero-meaned before quantization and have scaling factors.
- in Qin et al., 2021, the weights are both zero-meaned and uni-normalized.
Can the authors specify in more detail the derivation of (2), and the connection/difference with the two papers?

The theory part also requires some more clarifications. Specifically,
- Theorem 2 is based on the assumption that B_Q and B_K are entropy maximized. However, it is not clear why this assumption holds.
- Even though the softmax function is order-preserving, maximizing the entropy of sign(A - \phi(\tau, A)) is not equivalent to the same as the original problem in (9).
- Moreover, given the implication from Theorem 2 that the optimal \phi(\tau) is zero,  the binarization of the attention scores  sign(A - \phi(\tau, A)) =  sign(A) does not equal bool(A) in equation (11).

The experiments also require some more clarification. According to the original papers of BinaryBERT and TernaryBERT, the activations are only quantized to as low as 4 bits. However, in Tables 2-3, activations of these two methods are quantized to 1 or 2 bits. How are the activations quantized for these two methods?

Minors:
- Figure 3: missing the full stop at the end of the caption.

**Summary Of The Paper:**

This paper proposes to binarize both the weights and activations of the BERT model. The authors find that binarizing MHA causes the most significant accuracy drop and simple distillation on MHA by minimizing the MSE between the attention scores of the full-precision model and the quantized model can harm the performance sometimes. Thus the authors propose new binary representations that maximize entropy and distillation methods that maintain direction. Empirical results on the GLUE benchmark show that the method achieves good performance
when the weights are binarized and the activation is binarized/ternarized.


**Summary Of The Review:**

This paper is overall structured clearly and easy to follow. However, the motivation, theory part and experiments require some further clarifications.

---

> ### Author Response · Authors · 2021-11-15
> **Response to Reviewer tpUQ (1/3)**
>
> We thank the reviewer for the feedback and comments. We respond to the concerns below:
>
> **Q1:** The authors stated that the bi-linear quantization function in Equation (2) is motivated by Rastegati et al., 2016 and Qin et al., 2021. However, (2) is quite different from the quantization methods in both papers. Specifically,
>
> - in Rastegati et al., 2016, both the weights and activations are not zero-meaned before quantization and have scaling factors.
> - in Qin et al., 2021, the weights are both zero-meaned and uni-normalized. Can the authors specify in more detail the derivation of (2), and the connection/difference with the two papers?
>
> **A1:** We are sorry for your confusion. For the bi-linear defined in Eq. (2), we follow XNOR-Net [1] to apply the scaling factors calculated from the mean of the absolute value to minimize the quantization error. The weight balance technique is also applied in XNOR-Net according to their official implementation though it is not mentioned in their paper (please refer to https://github.com/allenai/XNOR-Net/blob/4006d594fe0349a5ed4a5ba6795641d9eecb3aac/train.lua#L169 ).  And [2] first points out that the weight balance technique improves the binarized network by maximizing the information entropy of binarized parameters, which presents 1.3% gain (85.2% vs 86.5% [2]) on ResNet-20 architecture and CIFAR-10 dataset. Furthermore, in Table A1.1, we further evaluate that the bi-linear with zero-mean weight on some tasks that 1-1-1 BinaryBERT crashed (CoLA and RTE), which shows that it significantly improves the accuracy and allows the model to converge.
>
> **Table A1.1: Comparison results on CoLA and RTE**
>
> | Method                 | #Bit  | CoLA | RTE   |
> | ---------------------- | ----- | ---- | ----- |
> | BinaryBERT             | 1-1-1 | 0.0% | 52.7% |
> | BinaryBERT (bi-linear) | 1-1-1 | 8.3% | 54.2% |
>
> The difference between the bi-linear we applied and XNOR-Net is that the scaling factor of activation is not used because it is dynamically calculated in real-time during inference which increases the computational cost. This is also a common practice in many recent binarization works [2,3,4].
>
>
>
> **Q2:** Theorem 2 is based on the assumption that B_Q and B_K are entropy maximized. However, it is not clear why this assumption holds.
>
> **A2:** Since the application of the bi-linear defined in Eq. (2) (and discussed in A1), the expected information entropy of both B_Q and B_K is maximized.
>
> As shown in Eq. (3) and Eq. (4), Q and K are output by the corresponding bi-lieanr and then binarized by the sign function to obtain $\mathbf{B_Q}$ and $\mathbf{B_K}$. The application of zero-mean pre-binarized weight in the bi-linear layer maximizes the information entropy of the binarized activation (such as $\mathbf{B_Q}$ and $\mathbf{B_K}$) in fully binarized BERT as described in Section 3.2.1 (Paragraph 1), and the related proof is shown in Appendix A.1.

---

> > ### Author Response · Authors · 2021-11-15
> > **Response to Reviewer tpUQ (2/3)**
> >
> > **Q3:** Even though the softmax function is order-preserving, maximizing the entropy of sign(A - \phi(\tau, A)) is not equivalent to the same as the original problem in (9).
> >
> > **A3:** We clarify that when the elements in tensor $\mathbf A$ are assumed as independent and identically distributed, there exists a $\phi(\tau, \mathbf A))$ making the entropy maximization of $\operatorname{sign}(\mathbf A - \phi(\tau, \mathbf A))$ be equivalent to that of $\operatorname{sign}(\operatorname{softmax}(\mathbf A) - \tau)$.
> >
> > Specifically, considering the order-preserving characteristic of both $\operatorname{sign}$ and $\operatorname{softmax}$, only the largest $n$ elements (the value of $n$ is related to the variable $k$ for specific matrix $\mathbf A$ in Theorem 1) are binarized to $1$ while others are binarized to $-1$. Therefore, when $\operatorname{sign}(\operatorname{softmax}(\mathbf A) - \tau)$ is optimized to have the maximized information entropy, there exists a corresponding threshold $\phi(\tau, \mathbf A)$ that maximizes the information entropy of $\operatorname{sign}(\mathbf A-\phi(\tau, \mathbf A))$. In addition, the conclusion of Theorem 2 that $\mathbf A\sim\mathcal N(0, D)$ suggests that a fixed threshold $\phi(\tau, \mathbf A)=0$ maximizes the information entropy of the binarized attention weight.
> >
> > **Proof.** As shown in Eq. (8), the information entropy of $\mathbf{\hat{B}_{\mathbf{w}}}^A =\operatorname{sign}(\operatorname{softmax}(\mathbf{A})-\tau)$ can be expressed as:
> >
> > $$\mathcal H(\mathbf{\hat{B}_{\mathbf{w}}}^A)=$$
> >
> > $$-\left(\int_{\tau}^{\infty} f(a^s) da^s\right) \log\left(\int_{\tau}^{\infty} f(a^s) da^s \right)-\left(\int_{-\infty}^{\tau} f(a^s) da^s\right) \log\left(\int_{-\infty}^{\tau} f(a^s) da^s\right),$$
> >
> > where $a^s = \operatorname{softmax}_A(a)$ and $a \in \mathbf A$. Since $\operatorname{softmax}$ function is strictly order-preserving, there exists a $\phi(\tau, \mathbf A)$ makes $\forall a>\tau$, $a^s> \phi(\tau, \mathbf A)$, and it thereby makes following equations established:
> >
> > $$\int_{\tau}^{\infty} f(a^s) da^s=\int_{\phi(\tau, A)}^{\infty} g(a) da,$$
> >
> > $$\int_{-\infty}^{\tau} f(a^s) da^s=\int_{\infty}^{\phi(\tau, A)} g(a) da,$$
> >
> > where $f$ and $g$ are the probability density functions of variable $a^s$ and $a$, respectively. And the information entropy  is of $\mathbf{\hat{B}_{\mathbf{w}}}^A$ equal to that of $\mathbf{B_A}$, and $\mathbf{B_A} = \operatorname{sign}(\mathbf A-\phi(\tau, \mathbf A))$. Therefore, when $\tau$ is optimized to maximize information, the information of $\mathbf {B_A}$ is also maximized under the threshold $\phi(\tau, \mathbf A)$.
> >
> >
> >
> > **Q4:** Moreover, given the implication from Theorem 2 that the optimal \phi(\tau) is zero, the binarization of the attention scores sign(A - \phi(\tau, A)) = sign(A) does not equal bool(A) in equation (11).
> >
> > **A4:** We further clarify and prove that for binarization functions sign and bool, the conditions of information entropy maximum for binarized parameters are the same, and $\phi(\tau)=0$ in both cases. Based on this prerequisite, Bi-Attention applies the bool function to binarize attention weight for reviving the attention mechanism (in Section 3.2.2).
> >
> > Specifically, since our bool function only redefined the value after binarization but keep the threshold of binarization at 0 (same as the sign function), the information entropy of binarized parameters can be expressed as:
> >
> > $\max_{\phi}  \mathcal H(\operatorname{sign}(A-\phi)) = -(\int_{\phi}^{\infty} f(x)) \log (\int_{\phi}^{\infty} f(x)) - (\int_{-\infty}^{\phi} f(x)) \log (\int_{-\infty}^{\phi} f(x)),$    (1)
> >
> > $\max_{\phi} \mathcal H(\operatorname{bool}(A-\phi)) = -(\int_{\phi}^{\infty} f(x)) \log (\int_{\phi}^{\infty} f(x)) - (\int_{-\infty}^{\phi} f(x)) \log (\int_{-\infty}^{\phi} f(x)).$    (2)
> >
> > Since $\max_{\phi} \mathcal H(\operatorname{sign}(A-\phi))$ and $\max_{\phi} \mathcal H(\operatorname{bool}(A-\phi))$ are trivially equal, the optimal $\phi$ for (1) and (2) are the same. Theorem 2 further shows that an optimal value is 0.
> >
> > The application of the bool function to binarize attention weight in Bi-Attention is for reviving the attention mechanism to capture crucial elements, and the above discussion shows that using it instead of sign function does not cut down the information entropy which is statistical maximized. Moreover, we point out that since the elements of binarized attention weight in the fully binarized BERT baseline are all 1 (all values before binarization are greater than 0), the discussion on its attention weight applies to any case of sign and bool functions (the latter is applied in the Bi-Attention of BiBERT), no matter it is from aspects of the information entropy or the specific value.

---

> > > ### Author Response · Authors · 2021-11-15
> > > **Response to Reviewer tpUQ (3/3)**
> > >
> > > **Q5:** The experiments also require some more clarification. According to the original papers of BinaryBERT and TernaryBERT, the activations are only quantized to as low as 4 bits. However, in Tables 2-3, activations of these two methods are quantized to 1 or 2 bits. How are the activations quantized for these two methods?
> > >
> > > **A5:** Thanks for pointing out that. We have updated more details of BinaryBERT [5] and TernaryBERT [6] under 1-bit and 2-bit activation settings in the revision (Appendix B.2).
> > >
> > > BiBERT is the first work towards the fully binarized BERT with 1-bit weight, activation, and embedding, and enjoys extremely high computational efficiency and memory usage as we discussed in A1. To be fair, all the networks in the comparative experiments should be set to the same 1-1-1 bit-width to have a similar computation and storage level, otherwise, the accuracy comparison does not make sense. However, there is no existing BERT quantization method that binarizes activation, we thereby have to use the most closely related works BinaryBERT and TernaryBERT as the major competitors, which quantize weight and embedding to ultra-low bit-width (1-bit or 2-bit), and further reduce the activation to 1-bit or 2-bit to compare the accuracy rate under similar computational and storage efficiency. The quantization processes of activation follow official or fair implementation as much as possible.
> > >
> > > For BinaryBERT, except 4-bit and 8-bit activation settings mentioned in their paper [5], the official code provides the implementations for 1-bit or 2-bit activation settings, which are Binary-Weight-Network (BWN) and Ternary-Weight-Network (TWN), respectively, and we completely follow the released code.
> > >
> > > Specifically, we apply Binary-Weight-Network (BWN) and Ternary-Weight-Network (TWN) methods to activation under 1-bit and 2-bit input settings, respectively. The formulations and implementations are presented in [5] and official codes, respectively:
> > >
> > > BWN: $\mathcal{Q}(a_{i}^{b})=\beta \cdot \operatorname{sign}\left(a_{i}^{b}\right), \beta=\frac{1}{n}\left\|\mathbf{a}^{b}\right\|_{1},$
> > >
> > > TWN: $\mathcal{Q}\left(a_{i}^{t}\right)=\beta \cdot \operatorname{sign}\left(a_{i}^{t}\right), \left|a_{i}^{t}\right| \geq \Delta; 0,  \left|a_{i}^{t}\right|<\Delta.$
> > >
> > > BinaryBERT 1-bit & 2-bit input: [https://github.com/huawei-noah/Pretrained-Language-Model/blob/1dbaf58f17b8eea873d76aa388a6b0534b9ccdec/BinaryBERT/transformer/utils_quant.py](https://github.com/huawei-noah/Pretrained-Language-Model/blob/1dbaf58f17b8eea873d76aa388a6b0534b9ccdec/BinaryBERT/transformer/utils_quant.py#L428) (in the act_quant_fn function)
> > >
> > > For TernaryBERT, since there is no recommended setting under 1-bit and 2-bit input in its official code, we use the same implementation as BinaryBERT to quantize activation (BWN for 1-bit and TWN for 2-bit) for fair comparison.
> > >
> > > In fact, existing work [5] has also pointed out that quantizing the activation is of great challenge and the ultra-low bit-width activation quantization on the existing method may lead to too bad results, which is exactly the problem our BiBERT devotes to solve.  Compared with TernaryBERT and BinaryBERT, under the 1-bit input setting, our BiBERT applies a simple sign function to binarize activation, while the former two apply real-time calculated scaling factors. Therefore, benefitting from the well-designed structure and distillation scheme, our BiBERT achieved better results with a smaller amount of computation.
> > >
> > >
> > >
> > > **Q6:** Minors: Figure 3: missing the full stop at the end of the caption.
> > >
> > > **A6:** Thanks for pointing out. We revised the issue in the revision and carefully revised the full manuscript.
> > >
> > >
> > >
> > > **Reference**
> > >
> > > [1] Rastegari, Mohammad, et al. "Xnor-net: Imagenet classification using binary convolutional neural networks." ECCV. 2016.
> > >
> > > [2] Qin, Haotong, et al. "Forward and backward information retention for accurate binary neural networks." CVPR. 2020.
> > >
> > > [3] Liu, Zechun, et al. "Bi-real net: Enhancing the performance of 1-bit cnns with improved representational capability and advanced training algorithm." ECCV. 2018.
> > >
> > > [4] Liu, Zechun, et al. "Reactnet: Towards precise binary neural network with generalized activation functions." ECCV. 2020.
> > >
> > > [5] Bai, Haoli, et al. "BinaryBERT: Pushing the Limit of BERT Quantization." ACL. 2021.
> > >
> > > [6] Zhang, Wei, et al. "TernaryBERT: Distillation-aware Ultra-low Bit BERT." EMNLP. 2020.

---

> > > > ### Comment · Reviewer_tpUQ · 2021-11-29
> > > > **Response to Authors**
> > > >
> > > > I thank the authors for the detailed feedback. However, my concerns regarding the connection with XNOR-Net and fair comparison with BinaryBERT and TernaryBERT still exist. Specifically, (1) though XNOR-Net uses its weight balance in its implementation, the scaling should be calculated based on the zero-meaned weights. (2) I checked the official implementation of BianryBERT https://github.com/huawei-noah/Pretrained-Language-Model/blob/1dbaf58f17b8eea873d76aa388a6b0534b9ccdec/BinaryBERT/transformer/utils_quant.py#L434. The authors clearly commented on the code they do not suggest using BWN or TWN for activation quantization. Instead, for 2-bit activations, they have options of using uniform quantization or LSQ. For fair comparison with BinaryBERT, the authors should use the suggested 2-bit activation quantization. Otherwise it is hard to tell whether the proposed method still outperforms BinaryBERT under a fair setting.

---

> > > > > ### Author Response · Authors · 2021-11-29
> > > > > **Re: Response to Authors (1/2)**
> > > > >
> > > > > Thank you for your feedback and suggestions.
> > > > >
> > > > > **Q1:** Concerns regarding the connection with XNOR-Net: Although XNOR-Net uses its weight balance in its implementation, the scaling should be calculated based on the zero mean weight.
> > > > >
> > > > > **A1:** Thank you for pointing it out. Compared with XNOR-Net, the bi-linear in our BiBERT applies the same weight balance, while some differences still exist: (1) the scaling factor of activation in bi-linear is removed for the computational efficiency since it will cause an up to 0.3 GFLOPs (43%) computation increment in fully binarized BERTs; (2) the other minor difference is that bi-linear calculates the scaling factor before the weight balance (while XNOR-Net calculates it afterward). When building the baseline, we found that calculating the scaling factor for weight before balance usually leads to better results (e.g., 11.7% vs. 11.1% for baselines on CoLA). The benefits seem to come from more significant feature heterogeneity brought by larger values of scaling factors, which improves fully binarized BERTs in mitigating information degradation. We thus apply this setting to build all fully binarized BERTs. However, whether calculating scaling factors before or after weight balance does not affect the theory part of BiBERT. And the experiments in Table RA1.1 also show little difference in accuracy (BiBERT 61.0% vs. BiBERT$_\text{scale}$ 60.9% on average). Some of our designs in BiBERT may eliminate the influence of rearranging the calculation. We will show more results and related discussions in the new revision.
> > > > >
> > > > > In addition, the improvement brought by bi-linear mainly depends on the weight balance that maximizes the information of binarized activation and weight, which is discussed in A1 of the original response (1/3) and also practiced in XNOR-Net and IR-Net.
> > > > >
> > > > > **Table RA1.1 Comparison results without data augmentation.**
> > > > >
> > > > > | Quant                           | #Bits | FLOPs(G)        | MNLI          | QQP      | QNLI     | SST-2    | CoLA     | STS-B    | MRPC     | RTE      | Avg.     |
> > > > > | ------------------------------- | ----- | --------------- | ------------- | -------- | -------- | -------- | -------- | -------- | -------- | -------- | -------- |
> > > > > | Baseline                        | 1-1-1 | 0.4             | 45.8/47.0     | 73.2     | 66.4     | 77.6     | 11.7     | 7.6      | 70.2     | 54.1     | 50.4     |
> > > > > | BinaryBERT                      | 1-1-1 | 0.4             | 35.6/35.3     | 63.1     | 51.5     | 53.2     | 0        | 6.1      | 68.3     | 52.7     | 40.6     |
> > > > > | BinaryBERT$_\text{TWN}$ (paper) | 1-1-2 | 0.8 ($2\times$) | 35.4/35.2     | 63.1     | 52.6     | 82.5     | 14.6     | 6.5      | 68.3     | 52.7     | 45.7     |
> > > > > | BinaryBERT$_\text{LSQ}$         | 1-1-2 | 0.8 ($2\times$) | 35.9/35.7     | 63.1     | 53.1     | 83.0     | 14.2     | 6.8      | 68.3     | 52.7     | 45.9     |
> > > > > | BinaryBERT$_\text{uniform}$     | 1-1-2 | 0.8 ($2\times$) | 35.9/35.5     | 63.1     | 52.5     | 82.1     | 14.5     | 6.1      | 68.3     | 52.7     | 45.6     |
> > > > > | BiBERT (paper)                  | 1-1-1 | **0.4**         | **59.3/60.0** | 82.4     | 70.2     | 86.9     | **25.3** | **33.5** | **72.9** | **58.5** | **61.0** |
> > > > > | BiBERT$_\text{scale}$           | 1-1-1 | **0.4**         | 59.1/59.8     | **82.7** | **70.3** | **87.0** | 25.1     | 33.1     | 72.7     | 58.4     | 60.9     |

---

> > > > > > ### Author Response · Authors · 2021-11-29
> > > > > > **Re: Response to Authors (2/2)**
> > > > > >
> > > > > > **Q2:** Concerns regarding the fair comparison with BinaryBERT and TernaryBERT: I checked the official implementation of BinaryBERT https://github.com/huawei-noah/Pretrained-Language-Model/blob/1dbaf58f17b8eea873d76aa388a6b0534b9ccdec/BinaryBERT/transformer/utils_quant.py#L434. The author clearly commented that they do not recommend using BWN or TWN for activation quantification codes. Instead, for 2-bit activation, they can choose to use unified quantization or LSQ. In order to make a fair comparison with BinaryBERT, the author should use the recommended 2-bit activation quantification. Otherwise, it is difficult to judge whether the proposed method is still better than BinaryBERT under a fair setting.
> > > > > >
> > > > > > **A2:** Thank you for your constructive suggestions.
> > > > > >
> > > > > > We would like to first emphasize that BiBERT is the first fully binarized BERT with extremely high computational and storage savings. A completely fair comparison should be performed among models with the same bit width (1-1-1), however, the 1-1-1 models binarized by existing BERT quantization methods (such as BinaryBERT) suffer severe accuracy drops and even crash directly on many tasks. In contrast, 1-1-2 BinaryBERT has a computational cost of $2\times$, but 1-1-1 BiBERT still surpasses it in terms of accuracy.
> > > > > >
> > > > > > Moreover, we also present the results of 1-1-2 BinaryBERT obtained by uniform (L434) and LSQ (L438) methods in Table RA1.1 following the reviewer's suggestions. The accuracy of 1-1-2 BinaryBERT$_\text{uniform}$ and BinaryBERT$_\text{LSQ}$ is similar to that of BinaryBERT$_\text{TWN}$ (TWN 45.7% vs. uniform 45.6% & LSQ 45.9% on average). These results show that the bottleneck of 1-1-2 BinaryBERT performance is the limited representation capability caused by ultra-low bit-width activations, and also show that existing BERT quantization methods cannot be directly used to obtain BERTs with extremely low-bit activations (1-bit or 2-bit).

---

### Official Review · Reviewer_T1kK · 2021-11-02

**Correctness:** 4
**Technical Novelty And Significance:** 3
**Empirical Novelty And Significance:** 2
**Recommendation:** 6
**Confidence:** 3

**Main Review:**

Strength:

The idea is well-motivated and easy to follow. The paper starts from information theory and focuses on the mutual information between binarized and full-precision representations. Considering that the ideal binarized representation should preserve the given full-precision counterparts as much as possible, it is very natural to maximize the mutual information between two representations.

The authors also find that the direction mismatch hinders the accurate optimization of the fully binarized BERT. This finding is interesting and well-motivated.

Experiment results show that the proposed approaches achieve much better results than previous baselines.

Weakness:

 I have noticed that [1] reported higher results in their paper.  It would be better to explain why the higher results are not reported in this paper.

The work of BinaryBERT uses 1-bit weight quantization and 4-bit activation quantization. It not only achieves better results than the proposed approach,  but also almost reaches the results of the full-precision model.  Since BinaryBERT and BiBERT have similar parameter sizes, the empirical contribution of BiBERT is a little bit marginal to me.

[1] BinaryBERT: Pushing the Limit of BERT Quantization


**Summary Of The Paper:**

This paper focuses on the full binarization of BERT. Since current pre-trained models achieve promising results on NLP tasks, it is a trend to apply these models to real-world applications with constraint computation resources.  Quantization is one of the prime choices to reduce computation requirements. In recent years, BERT quantization has made significant progress. However, training binary BERT is still an important but challenging question. This paper finds that the worse performance of binary BERT can be mainly attributed to information degradation and optimization direction mismatch respectively in the forward and backward propagation. To address these problems, the authors propose two solutions, Bi-Attention and Direction-Matching Distillation. Experimental results on GLUE benchmark demonstrate that the proposed work outperforms several popular baselines. Also, ablation studies verify the effectiveness of the proposed solutions.

**Summary Of The Review:**

It is a well-motivated paper presenting an alternative understanding of the challenges of training binarized BERT. The empirical contribution is a little bit marginal since there is a large gap between the proposed approach and the best result in [1].

---

> ### Author Response · Authors · 2021-11-15
> **Response to Reviewer T1kK**
>
> We would like to first express our deep appreciation for your comments, and our response are as follows:
>
>
>
> **Q1:** I have noticed that [1] reported higher results in their paper. It would be better to explain why the higher results are not reported in this paper.
>
> The work of BinaryBERT uses 1-bit weight quantization and 4-bit activation quantization. It not only achieves better results than the proposed approach, but also almost reaches the results of the full-precision model. Since BinaryBERT and BiBERT have similar parameter sizes, the empirical contribution of BiBERT is a little bit marginal to me.
>
> BinaryBERT: Pushing the Limit of BERT Quantization
>
> **A1:** We first need to state that the fully binarized network with binarized weight and activation (and embedding) is completely different from the binarized weight network since the former far surpasses the latter in computation and energy efficiency [1,2]. Fully binarized parameters bring extremely efficient XNOR-Bitcount bitwise operations [2] with ultra-low energy consumption (achieve ${89\times}$ energy efficiency) [3] and lower storage usage. Therefore, the full binarization allows the network to be compressed and deployed on limited computing resources and ultra-low power devices, such as mobile phones and cameras, and also be suitable for applications that require real-time interaction and fast response. However, the extremely limited representation capabilities and the difficult discrete optimization make the accurate fully binarized networks a great challenge. As our experiments show, when we push the other existing methods to fully binarized (such as BinaryBERT [4]), the accuracy of these models drops sharply and does not even converge on some tasks.
>
> The BiBERT is the first full binarization method for BERT models that applies 1-bit weights, activations, and embeddings, and thereby saves extreme  ${56.3\times}$ computational FLOPs and ${31.2\times}$ storage compared to the full-precision BERT. As a closely related competitor, BinaryBERT [4] only binarizes the weight and embedding while keeping activation to 4-bit or 8-bit, which results in executing the more computation-consuming integer operation rather than bitwise operation. For example, 1-1-4 BinaryBERT suffers ${3.8\times}$ FLOPs consumption compared to 1-1-1 BiBERT. Therefore, BiBERT has a huge improvement in computing efficiency.
>
> To be fair, all the networks in the comparative experiments should be set to the same 1-1-1 bit-width to have a similar computation and storage level, otherwise, the accuracy comparison does not make sense. For BinaryBERT, the 4-bit activation setting brings a stronger representation with ${3.8\times}$ computational consumption than BiBERT, which reasonably makes the accuracy of the network far exceed that with 1-bit activation. Our BiBERT not only surpasses all existing methods at the 1-1-1 bit setting but also surpasses some of them with higher activation bit-widths, such as 1-1-2 bit BinaryBERT (${2.0\times}$ FLOPs consumption) and 2-2-2 bit TernaryBERT (${3.8\times}$ FLOPs consumption), which more forcefully proves that our designs help the full binarized BERT to achieve higher accuracy.
>
>
>
> **Reference**
>
> [1] Courbariaux, Matthieu, Yoshua Bengio, and Jean-Pierre David. "Binaryconnect: Training deep neural networks with binary weights during propagations." NeurIPS. 2015.
>
> [2] Rastegari, Mohammad, et al. "Xnor-net: Imagenet classification using binary convolutional neural networks." ECCV. 2016.
>
> [3] Chen, Gang, et al. "PhoneBit: efficient gpu-accelerated binary neural network inference engine for mobile phones." DATE. 2020.
>
> [4] Bai, Haoli, et al. "BinaryBERT: Pushing the Limit of BERT Quantization." ACL. 2021.

---

### Official Review · Reviewer_yAoP · 2021-11-03

**Correctness:** 4
**Technical Novelty And Significance:** 3
**Empirical Novelty And Significance:** 3
**Recommendation:** 8
**Confidence:** 3

**Main Review:**

The paper proposes a full binarization of BERT (including 1-bit activation), which is a very challenging task. The proposed bi-attention for maximum information entropy is a very interesting design for fully binary attention models. My question here is in section 2.1, when analyzing the effect of each distillation term, whether the authors used a fully binarized attention map which only contains 1, just like the Figure 4(b). In that case, I don't understand the meaning of attention distillation in Figure 3 since the binary attention map is fixed.

It is also interesting to see that the activation distillation can cause the optimization direction when the quantization becomes more aggressive, which well motivates the proposed direction-matching distillation term.

The experimental results are solid and impressive. In the 1-1-1 setting in which the paper mainly focused, the proposed BiBERT model achieves significant improvement over the 1-1-1 baseline model.

**Summary Of The Paper:**

This paper proposes a full binarization of BERT (including 1-bit activation), called BiBERT. The key observation of the paper is that binary activation becomes a big challenge for the attention weights in self-attention. To solve this problem, the paper proposes a new attention binarization approach that maximizes the information entropy. The paper further proposes a Direction-Matching Distillation (DMD) scheme which encourages the similarity pattern matrices of keys, values, and hidden states in the original BERT and BiBERT to be minimized. Experimental results on GLUE datasets show that BiBERT significantly outperforms other models under the full binarization scheme.

**Summary Of The Review:**

The paper is well-motivated, novel, and well supported by the experiments.

---

> ### Author Response · Authors · 2021-11-15
> **Response to Reviewer yAoP**
>
> We are deeply grateful for the reviewer’s support of our work and we thank the reviewer for the constructive and helpful suggestions. We provide additional discussions below:
>
> **Q1:** My question here is in section 2.1, when analyzing the effect of each distillation term, whether the authors used a fully binarized attention map which only contains 1, just like the Figure 4(b). In that case, I don't understand the meaning of attention distillation in Figure 3 since the binary attention map is fixed.
>
> **A1:** We clarify that, as in Eq. (6) presents, the fully binarized BERT baseline distills the full-precision attention scores $\mathbf{A}$ obtained by Eq. (4) instead of the binarized attention weight (attention map) $\mathbf{B_A^s}$ (or $\mathbf{B_A}$). Thus, though the forward representation of the binarized attention weight is fixed as shown in Figure 4, the distillation loss of the attention score still updates the student network through backward propagation.
>
> In the fully binarized BERT baseline introduced in Section 2, the distillation loss of the attention score is adopted following the settings in [1,2]. The original intention of this design is to enable the quantized student network to utilize the knowledge in the attention scores of the full-precision teacher BERT. However, our observation experiments show that the distillation of attention score instead hinders the training of the student BERT in the full binarization (as shown in Figure 3(b)), which is further verified in Section 3 to be caused by the optimization direction mismatch in the complete binarization of the attention structure. This phenomenon motivates us to design the novel Direction-Matching Distillation (DMD) scheme for fully binarized BERT, which distills the upstream query $\mathbf Q$ and key $\mathbf K$ instead of attention score in our DMD to utilize its knowledge and alleviate direction mismatch. The experiments show that DMD provides the matching optimization direction (Figure 5(b)) and significantly improves the performance.
>
> **Reference**
>
> [1] Zhang, Wei, et al. "TernaryBERT: Distillation-aware Ultra-low Bit BERT." EMNLP. 2020.
>
> [2] Jiao, Xiaoqi, et al. "TinyBERT: Distilling BERT for Natural Language Understanding." EMNLP Findings. 2020.

---

> > ### Comment · Reviewer_yAoP · 2021-11-19
> > **Response to the Authors**
> >
> > I have read the other reviews and the authors' responses. In fact, the author's response does not address my concerns on the baseline binarized BERT.
> >
> > From the authors' response, it seems to me that all the baseline methods (including the ones that this paper converts into binarized activation versions) use a fixed all one attention map. Such kind of map is just an average pooling of all tokens and would be meaningless to be adopted as an attention map. I'm afraid this would result in under-estimating the performance of vanilla fully binarized BERT and previous approaches in the fully binarized setting.
> >
> > I'm going to temporarily decrease my score until the authors can provide comparisons to baselines with binarized attention in more meaningful settings (for example, directly use a threshold to control 50% (or 30% / 70%) elements in the attention map to be zeros. Since this paper is the first paper to develop a fully binarized version of BERT, setting up good baselines would be very important.

---

> > > ### Author Response · Authors · 2021-11-22
> > > **Re: Response to Authors (1/3)**
> > >
> > > **Q1:** From the authors' response, it seems to me that all the baseline methods (including the ones that this paper converts into binarized activation versions) use a fixed all one attention map. Such kind of map is just an average pooling of all tokens and would be meaningless to be adopted as an attention map. I'm afraid this would result in under-estimating the performance of vanilla fully binarized BERT and previous approaches in the fully binarized setting.
> > >
> > > I'm going to temporarily decrease my score until the authors can provide comparisons to baselines with binarized attention in more meaningful settings (for example, directly use a threshold to control 50% (or 30% / 70%) elements in the attention map to be zeros. Since this paper is the first paper to develop a fully binarized version of BERT, setting up good baselines would be very important.
> > >
> > >
> > >
> > > **A1:** We thank you for your constructive feedback. We compare and discuss more solutions that diversify the values of binarized weights (instead of the all one attention map), including the baselines with 50% zero binarized attention weight, with the asymmetric quantization function, and with mean value threshold. We add the results and related discussion in Section 4.2 and Appendix C.1 in our revision.
> > >
> > > ## **1. Reasons of building the original baseline**
> > >
> > > When we built the fully binarized BERT baseline, we considered the following reasons and chose the original baseline settings shown in the paper:
> > >
> > > (1) Additional inference computation should be avoided to the greatest extent compared with the BERTs obtained by direct full binarization. We discard the scaling factor for activation in the bi-linear unit since it requires real-time updating during inference and increases floating-point operations. In the attention structure, we directly binarize the activation without re-scaling or mean-shifting.
> > >
> > > (2) Since there is no previous work to fully binarize BERT, the existing representative binarization techniques are chosen to build the baseline. The binarization function with a fixed 0 threshold is applied to the original definition of the binarized neural network [1] and is used by default in most binarization works [2,3,4], we thereby initially apply this function when building the fully binarized BERT baseline. Besides, it also helps to more objectively evaluate the benefits of maximizing information entropy in the attention structure.
> > >
> > > Therefore, the fully binarized baseline under the current BERT architecture applies the representative techniques and almost without additional floating-point computation.
> > >
> > > ## **2. Different solutions for attention weight**
> > >
> > > Causing by the application of softmax, the attention possibilities in the attention structure follow the probability distribution and (attention weight) are all one after binarization. In fact, we were aware of the problem when building the fully binarized baseline, and had evaluated two solutions by existing quantization techniques.
> > >
> > > The first solution is to degenerate the asymmetric quantization function (usually applied to 2-8 bit quantization [5,6]) to the 1-bit case, since the method is originally designed to deal with the imbalance of value distribution in quantization:
> > >
> > > **Solution 1** (Baseline$_\text{aysm}$):
> > >
> > > $Q(x)=\max(x), x\ge0.5(\max(x)+\min(x)); \min(x), \text{otherwise}.$  (A1.1)
> > >
> > > The other solution is to simply use the mean of the elements as the threshold [1,2]:
> > >
> > > **Solution 2** (Baseline$_{\mu}$):
> > >
> > > $Q(x)=\operatorname{bool}(\mathbf x-\tau), \quad \tau = \mu(\mathbf x)$,  (A1.2)
> > >
> > > where $\mu(\cdot)$ denotes the mean value.
> > >
> > > We also thank the reviewers for providing the Solution 3, which is to use the threshold limiting the zero attention weight to a certain percentage (such as 50%):
> > >
> > > **Solution 3** (Baseline$_{0.5}$):
> > >
> > > $Q(\mathbf x)=\operatorname{bool}(\mathbf x-\tau), \quad \tau = Q_{0.5}(\mathbf x)$,  (A1.3)
> > >
> > > where $Q_{0.5}(\cdot)$ denotes the quantile of 50% percentage (median).
> > >
> > > The above three solutions are all able to alleviate the problem of fixed all one attention weight. We present the results of these solutions in Table A1.1 and further discuss them in detail.

---

> > > > ### Author Response · Authors · 2021-11-22
> > > > **Re: Response to Authors (2/3)**
> > > >
> > > > Table A1.1 Comparison results without data augmentation.
> > > >
> > > > | Solution  | Quant                  | #Bits | $\Delta$FLOPs(G) | MNLI          | QQP      | QNLI     | SST-2    | CoLA     | STS-B    | MRPC     | RTE      | Avg.     |
> > > > | --------- | ---------------------- | ----- | ---------------- | ------------- | -------- | -------- | -------- | -------- | -------- | -------- | -------- | -------- |
> > > > | Original  | Baseline               | 1-1-1 | 0                | 45.8/47.0     | 73.2     | 66.4     | 77.6     | 11.7     | 7.6      | 70.2     | 54.1     | 50.4     |
> > > > |           | BinaryBERT             | 1-1-1 | 0                | 35.6/35.3     | 63.1     | 51.5     | 53.2     | 0        | 6.1      | 68.3     | 52.7     | 40.6     |
> > > > | Solution1 | Baseline$_\text{asym}$ | 1-1-1 | 0.074 (15%)      | 45.1/46.3     | 72.9     | 64.3     | 72.8     | 4.6      | 9.8      | 68.3     | 53.1     | 48.6     |
> > > > | Solution2 | Baseline$_{\mu}$       | 1-1-1 | 0.076 (16%)      | 48.2/49.5     | 73.8     | 68.7     | 81.9     | 16.9     | 11.5     | 70.0     | 54.9     | 52.8     |
> > > > | Solution3 | Baseline$_{0.5}$      | 1-1-1 | 0.076 (16%)      | 47.7/49.1     | 74.1     | 67.9     | 80.0     | 14.0     | 11.5     | 69.8     | 54.5     | 52.1     |
> > > > |           | BinaryBERT$_{0.5}$    | 1-1-1 | 0.076 (16%)      | 39.2/40.0     | 66.7     | 59.5     | 54.1     | 4.3      | 6.8      | 68.3     | 53.4     | 43.5     |
> > > > | Ours      | BiBERT                 | 1-1-1 | **0**            | **59.3/60.0** | **82.4** | **70.2** | **86.9** | **25.3** | **33.5** | **72.9** | **58.5** | **61.0** |
> > > >
> > > > ### **2.1. Discussion of Solution 1**
> > > >
> > > > As shown in Eq. (A1.1), since the dynamically threshold ($0.5(\max(\mathbf x)+\min(\mathbf x))$) is applied in this quantizer, the elements of attention weight are quantized to $\max(\mathbf x)$ and $\min(\mathbf x)$ instead of a same value. However, the asymmetrical binarized values obtained by this quantizer make the binarized BERT hard to apply bitwise operations and lose the efficiency advantage brought by binarization (0.074G (15%) additional FLOPs), so that this practice is also not used in existing binarization works.
> > > >
> > > > And as the results show, the accuracy of the binarized BERT is also worse than the existing baseline (Baseline 50.4% vs. Baseline$_\text{asym}$ 48.6% on average, Row 2 and 4 in Table A1.1).
> > > >
> > > > ### **2.2. Discussion of Solution 2**
> > > >
> > > > As the effect of this solution on the weight parameter [1,2], it can ensure diversity of binarized attention weight elements instead of all 1 ($\min(\mathbf x) \le \tau \le \max(\mathbf x)$). But the premise of this practice is the Gaussian (symmetric) distribution of floating-point parameters before binarization [2], while the attention score follows an asymmetric probability distribution, causing the values of binarized activation to imbalanced.
> > > >
> > > > Therefore, the improvement of Baseline$_\mu$ in terms of accuracy is also limited (as shown in Row 2 and 5 in Table A1.1), and this solution also increases 0.076G (16%) computational FLOPs compared to the original baseline.

---

> > > > > ### Author Response · Authors · 2021-11-22
> > > > > **Re: Response to Authors (3/3)**
> > > > >
> > > > > ### **2.3. Discussion of Solution 3**
> > > > >
> > > > > The fixed 50% percentage of zero binarized attention weights suggested by the reviewer are good baseline settings that help to further clarify the entropy maximization motivation of the Bi-Attention in BiBERT. As shown in Table A1.1 (Row 6 and 7), the results of Baseline and BinaryBERT under this setting are significantly improved by 1.7% and 2.9% on average compare the original results (Row 2 and 3), respectively. And from the perspective of information entropy, a 50% percentage of zero attention weights can also ensure maximum information. We present the detailed ablation results (10%-90% zero attention weight) on STS-B in Table A1.2, which show the model maximizing information entropy achieves the best results.
> > > > >
> > > > > Table A1.2 Ablation results with 10%~90% zero attention weight on STS-B.
> > > > >
> > > > > | Percentage | 10%  | 30%  | 50%       | 70%   | 90%   |
> > > > > | ---------- | ---- | ---- | --------- | ----- | ----- |
> > > > > | Entropy    | 0.47 | 0.88 | **1.0**   | 0.88  | 0.47  |
> > > > > | Accuracy   | 8.5% | 8.6% | **11.5%** | 10.8% | 10.0% |
> > > > >
> > > > > Though forcing a certain ratio of zero attention weights improves the baseline, there exists shortcomings for fully binarized BERTs. First, the calculation of quantile thresholds relies on real-time computation or sorting, which increases computation of the fully binarized BERT in inference (about 0.076G (16%) additional FLOPs). Moreover, the practice of 50% quantile threshold should be regarded as the solution to a more stringent optimization problem (rather than the optimization problem in Eq. (9) of the paper) since it further constrains and maximizes the entropy of each tensor of binarized attention weight instead of optimizing the overall distribution. Thus, the quantile threshold is more restrictive for the binarized attention weights and limits their representation capability, causing the results of obtained fully binarized BERTs lower than those of models applying Bi-Attention.
> > > > >
> > > > > ## **3. Conclusion**
> > > > >
> > > > > Although the above three solutions improve accuracy to a certain extent, they only bring limited improvements while destroying the advantages of the fully binarized network on computational efficiency. As analyzed in our paper, the existing attention structure is not suitable for directly fully binarizing BERT, which is also the reason we specially designed the Bi-Attention structure for the fully binarized BERT. However, considering the reason mentioned by the reviewer that the fixed all one attention weight causes itself and related distillation to fail, and also to more comprehensively compare with our method, we add the results of these solutions (as strengthened baselines) to the paper as recommended by the reviewer.
> > > > >
> > > > >
> > > > >
> > > > >
> > > > > **Reference**
> > > > >
> > > > > [1] Rastegari, Mohammad, et al. "Xnor-net: Imagenet classification using binary convolutional neural networks." ECCV. 2016.
> > > > >
> > > > > [2] Qin, Haotong, et al. "Forward and backward information retention for accurate binary neural networks." CVPR. 2020.
> > > > >
> > > > > [3] Wang, Ziwei, et al. "BiDet: An efficient binarized object detector." CVPR. 2020.
> > > > >
> > > > > [4] Liu, Zechun, et al. "Reactnet: Towards precise binary neural network with generalized activation functions." ECCV. 2020.
> > > > >
> > > > > [5] Bai, Haoli, et al. "BinaryBERT: Pushing the Limit of BERT Quantization." ACL. 2021.
> > > > >
> > > > > [6] Zhang, Wei, et al. "TernaryBERT: Distillation-aware Ultra-low Bit BERT." EMNLP. 2020.

---

> ### Comment · Reviewer_yAoP · 2021-11-29
> **Response to Authors**
>
> I appreciate the authors' efforts in addressing my concerns. With the comparison to stronger baselines (instead of all-one attention), my main concern is addressed so I decide to raise my score back to 8.

---

### Comment · Area_Chair_3Z5s · 2021-11-14
**Additional Discussion Encouraged**

Dear Reviewers,

can you please take a look at each other's reviews? Your reviews currently straddle the decision boundary and it would be good to make sure you have considered all the perspectives provided. Please update your reviews (at least to acknowledge that you have read all reviews).

Thanks,
Your Area Chair

---

### Author Response · Authors · 2021-11-15
**General Response**

We are grateful for the ACs' and reviewers' positive feedback towards BiBERT. To assist a clearer understanding of our paper, we summarize our main contributions below:

We present BiBERT, the first accurate fully binarized BERT for efficient deep learning on NLP tasks, to alleviate the resource constraint for large pre-trained BERTs that run on resource-limited devices. Different from the existing BERTs with quantized integer parameters or with only binarized weight and embedding, fully binarized parameters of BiBERT bring extremely efficient XNOR-Bitcount bitwise operations and lower storage usage, which allows BERTs to be compressed and deployed on limited computing resources and ultra-low power devices, such as mobile phones and cameras. And applications that require real-time interaction and fast response can run steadily on these source-limited devices. However, the compact 1-bit parameters cause the fully binarized neural network to suffer extremely limited representation capabilities, and the optimization process is hindered by the discrete function, which makes the design of accurate fully binarized networks a great challenge.

In this paper, we identify that the severe performance drop of the fully binarized BERT baseline based on existing techniques can be mainly attributed to the information degradation and optimization direction mismatch respectively in the forward and backward propagation. To solve these problems, BiBERT introduces an efficient Bi-Attention structure for maximizing representation information statistically and a Direction-Matching Distillation (DMD) scheme to optimize the full binarized BERT accurately. BiBERT outperforms both the straightforward baseline and existing state-of-the-art quantized BERTs with ultra-low bit activations by convincing margins on the NLP benchmark. We highlight that as the first fully binarized BERT, our method yields impressive ${56.3\times}$ and ${31.2\times}$ saving on FLOPs and model size, demonstrating the vast advantages and potential of the fully binarized BERT model in real-world resource-constrained scenarios.

We also update our manuscripts; the change we made includes:

- In Section 4.2, we add and discuss the performance of BiBERT on compact architectures (TinyBERT-6L and TinyBERT-4L), which shows that our BiBERT has outstanding performance on various architectures;
- In Section 4.2, we add more discussions and analysis of related methods (especially BinaryBERT and TernaryBERT);
- In Appendix A.6, we add relevant discussions and proofs about the equivalence of information entropy maximization;
- In Appendix B.2, we add relevant details about the implementation of the comparison method (BinaryBERT and TernaryBERT) under 1-bit and 2-bit activation settings;
- In Section 4.2 and Appendix C.1, we add the results and related discussion of strengthened baselines.
- We carefully correct our references in the revised version.

For the detailed explanation, please see our responses to each reviewer.

---

### Public Comment · ~Jack_Ma3 · 2021-11-17
**About visualization of the results of proposed Bi-Attention structure**

Dear authors, thanks for your cool work. I'm really curious about the visualization of the attention weight obtained from the proposed Bi-Attention structure (Eq.11). Can you kindly show any examples if time permits？

---

### Decision · Program_Chairs · 2022-01-20

**Decision:**

Accept (Poster)

**Comment:**

This paper present a way to fully binarize a BERT model. The authors convincingly demonstrate that a naive binarization results in large quality losses and then propose amendments. It is pretty impressive that it is possible to get a fully binarized model to work at all.
At the same time, the quality losses are still significant and in practice one might prefer to use distillation (as long as the hardware doesn't require binarization). One could also envision combinations of the proposed technique (perhaps in the 1-1-4 setting) with distillation.